
**Dynamics of Variable Dusk-Dawn Flow Associated with Magnetotail**
**Current Sheet Flapping**
**James H. Lane[1], Adrian Grocott[1], Nathan A. Case[1], Maria-Theresia Walach[1]**
[1] Department of Physics, Lancaster University, Lancaster, UK
*Correspondence to:* James Lane (j.lane@lancaster.ac.uk)
**Abstract**
Previous observations have provided a clear indication that the dusk-dawn ($v_{\perp y}$) sense of
both slow (< 200 km s$^{-1}$) and fast (> 200 km s$^{-1}$) convective magnetotail flows is strongly
governed by the Interplanetary Magnetic Field (IMF) $B_y$ conditions. The related 'untwisting
hypothesis' of magnetotail dynamics is commonly invoked to explain this dependence, in
terms of a large-scale magnetospheric asymmetry. In the current study, we present Cluster
spacecraft observations from 12 October 2006 of earthward convective magnetotail plasma
flows whose dusk-dawn sense disagrees with the untwisting hypothesis of IMF $B_y$ control of
the magnetotail flows. During this interval, observations of the upstream solar wind
conditions from OMNI, and ionospheric convection data using SuperDARN, indicate a large-
scale magnetospheric morphology consistent with positive IMF $B_y$ penetration into the
magnetotail. Inspection of the in-situ Cluster magnetic field data reveals a flapping of the
magnetotail current sheet; a phenomenon known to influence dusk-dawn flow. Results
from the curlometer analysis technique suggest that the dusk-dawn flow perturbations may
have been driven by the $\boldsymbol{J} \times \boldsymbol{B}$ force associated with a dawnward-propagating flapping of
the magnetotail current sheet, locally overriding the expected IMF $B_y$ control of the flows.
We conclude that invocation of the untwisting hypothesis may be inappropriate when
interpreting intervals of dynamic magnetotail behaviour such as during current sheet
flapping.



## 1. Introduction

Convective magnetotail plasma flows at Earth, driven by the closing of magnetic flux via
reconnection as part of the Dungey Cycle (Dungey, 1961) have been studied extensively for
many years (e.g. Angelopoulos et al. 1992, 1994; Sergeev et al., 1996; Petrukovich et al.,
2001; Cao et al., 2006; McPherron et al., 2011; Frühauff & Glassmeier, 2016). Arguably, the
most well studied of these is the Bursty Bulk Flow (BBF). Angelopoulos et al. (1994) defined
BBFs as being channels of earthward plasma flow continually above 100 km s$^{-1}$, exceeding
400 km s$^{-1}$ at one point across some interval, usually across a timescale of a few minutes.
The flows are said to be the main transporter of mass, energy and flux in the magnetotail
(e.g. Angelopoulos et al., 1994; Nakamura et al., 2002; Grocott et al., 2004a; Kiehas et al.,
2018). Although their earthward nature is the key defining characteristic of BBFs, they will
invariably exhibit a dusk-dawn component in their bulk flow as well (e.g. Angelopoulos et
al., 1994; Petrukovich et al., 2001; Grocott et al., 2004b). Understanding the drivers of dusk-
dawn asymmetries in magnetospheric dynamics is an important element of geospace
research (e.g. Haaland et al., 2017).

A key factor that has been observed to influence the dusk-dawn direction of the
magnetotail flow is the $B_y$ component of the Interplanetary Magnetic Field (IMF). It is well
established that when the IMF reconnects with the dayside terrestrial magnetic field, a non-
zero IMF $B_y$ component leads to asymmetric loading of open flux into the polar cap (e.g.
Khurana et al., 1996; Tenfjord et al., 2015; Grocott et al., 2017; Ohma et al., 2019). This
results in a twisting of the magnetotail whereby the closed field lines are rotated about the
midnight meridian, and a $B_y$ component is superimposed onto the tail field as a
consequence of IMF $B_y$ penetration (Cowley, 1981; Petrukovich, 2011; Tenfjord et al., 2015).
Subsequently, following nightside reconnection, the tail will untwist (Grocott et al., 2004c),
with the excitation of multiple convective flow bursts, each with an earthward and dusk-
dawn component, in the tail and nightside ionosphere (Grocott et al., 2007). In order to be
consistent with the tail 'untwisting hypothesis', any convective flows associated with an
individual tail field line should share the same dusk-dawn direction (e.g. see Figure 3 of
Grocott et al., 2005). The role of IMF $B_y$ in the untwisting hypothesis has been examined
previously in a number of studies (e.g. Grocott et al, 2007; Pitkänen et al., 2013, 2015,
2017). These studies revealed that under prolonged positive IMF $B_y$ conditions, the
earthward flows are expected to exhibit a dawnward component in the northern
hemisphere ($B_x > 0$) and a duskward component in the southern hemisphere ($B_x < 0$), with
the opposite correlation for negative IMF $B_y$ conditions. IMF $B_y$ has been shown to govern
the dusk-dawn nature of these flows both during periods of steadier, slower convection
(Pitkänen et al., 2019), as well as during more transient, dynamic BBF-like intervals (Grocott
et al., 2007; Pitkänen et al., 2013). In the present study, we present observations of
dawnward and duskward directed flows that do not match this expected dependence on
IMF $B_y$, implying that the untwisting hypothesis is insufficient in this case. Instead, we
suggest that the flows are being driven by local perturbations due to dynamic behaviour of
the tail that are associated with flapping of the current sheet.

The current sheet, or 'neutral' sheet, lies in the equatorial plane at the center of the tail
plasma sheet and separates the earthward ($B_x > 0$) and tailward ($B_x < 0$) directed field (Ness,
1965). The current sheet is a highly dynamic region of the Earth's magnetotail which can
undergo various types of net motion, such as tilting due to lobe magnetic pressures (Cowley
et al., 1981; Tenfjord et al., 2017) as well as flapping. Flapping of the current sheet can
generally be described as a sinusoidal-like variation in $B_x$ of up to tens of nanoTesla, where
an observing spacecraft often measures repeated changes in the sign of $B_x$ (e.g. Runov et al.,
2009), indicative of crossings of the current sheet, with characteristic times ranging from a
few seconds to (more commonly) several minutes (e.g. Runov et al., 2009; Wu et al., 2016;
Wei et al., 2019). Drivers of current sheet flapping have been widely investigated, with
possible causes ranging from external solar wind/IMF changes (Runov et al., 2009),
induction of hemispheric plasma asymmetries (Malova et al., 2007; Wei et al., 2015), fast
earthward flow (Nakamura et al., 2009) as well as periodical, unsteady magnetotail
reconnection (Wei et al., 2019). Studies such as Volwerk et al. (2008) and Kubyshkina et al.
(2014) have illustrated that flapping of the current sheet can be associated with variable
dusk-dawn flow, potentially overriding any IMF $B_y$ control of the flow.

In this paper we present Cluster spacecraft observations of an interval of dynamic
magnetotail behaviour on 12 October 2006. Throughout this interval, Cluster 1 observed





oscillations in the magnetic field $B_x$ component, which we attribute to current sheet
flapping, concurrent with a series of convective fast flows with significant and variable dusk-
dawn components. The $B_y$ component of the concurrent upstream IMF had been largely
positive for several hours prior to the flapping. Consequently, the interval discussed here
provides an opportunity to investigate the possible competition of two distinct mechanisms
for control of the dusk-dawn flow: 1) IMF $B_y$ and 2) localized dynamics related to the
flapping of the current sheet. In contrast to studies which have come before such as those
presented by Grocott et al. (2007) and Pitkänen et al. (2015), the observed dusk-dawn
direction of transient flow enhancements in this case disagrees with that which might be
expected from the prevailing IMF $B_y$ conditions, despite clear evidence for global
penetration of positive IMF $B_y$. We therefore suggest that flapping of the current sheet had
locally overridden the IMF $B_y$ control of the dusk-dawn flow observed by Cluster 1.

**2. Instrumentation and Data Sets**
*2.1. Spacecraft Data*
The magnetospheric observations presented in this case study were made by the Cluster
multi-spacecraft (C1-C4) constellation (Escoubet et al., 2001). We make use of the fluxgate
magnetometer (FGM) onboard the Cluster spacecraft to obtain magnetic field
measurements (Balogh et al., 2001), and obtain our bulk ion velocity data from the Hot Ion
Analyser (HIA) on C1 and C3 calculated as on-board moments (Rème et al., 1997). The
magnetic field data presented are 5 vectors-per-second (0.2s res) which have been 1s
median-averaged, with the velocity data presented having spin resolution of just over 4s.
Where these datasets have been combined to produce parameters such as the plasma beta
and field-perpendicular velocities, we have resampled both the magnetic field and plasma
data to 5s resolution. All data are presented in geocentric solar magnetospheric (GSM)
coordinates unless stated otherwise.

The interval of study in this paper occurred between 00:00 – 00:55 UT on 12 October 2006.
At 00:00 UT the Cluster spacecraft were located in the near-Earth magnetotail plasma sheet,
in the pre-midnight sector. C1 was located at (X = −14.7, Y = 6.0, Z = −1.2) $R_E$, C2 at (X =
−14.2, Y = 7.5, Z = −0.7) $R_E$, C3 at (X = −13.9, Y = 7.0, Z = −2.1) $R_E$, and C4 at (X = −13.2, Y = 6.2,
Z = −0.8) $R_E$. This is depicted in Figure 1a by the colored triangles, along with the respective



spacecraft trajectories, from 00:00 – 00:55 UT, by the solid lines. Figure 1b shows a zoomed-
out version of Figure 1a, which illustrates the location of the spacecraft with respect to the
Earth. Figure 1b also shows a traced modelled magnetic field line, achieved using the semi-
empirical TA15 model of the magnetosphere (Tsyganenko & Andreeva, 2015), which passes
through the location of C1 and connects to both the northern and southern hemispheres of
the Earth. We parameterised the TA15 model using mean-averaged solar wind dynamic
pressure ($P_{dyn}$), IMF $B_y$ and IMF $B_z$ data from the 1-hour interval prior to 00:28 UT (the start
of our specific interval of interest). These values were $P_{dyn}$ = 1.56 nPa, IMF $B_y$ = +1.56 nT and
IMF $B_z$ = -2.17 nT. There was also a tailward dipole tilt of $\approx$-12$^\circ$. The model was also
parameterised with a solar wind coupling function index known as the 'N index', after
Newell et al. (2007). The N index varies between 0 (quiet) and 2 (very active), and in this
instance was ~0.4.

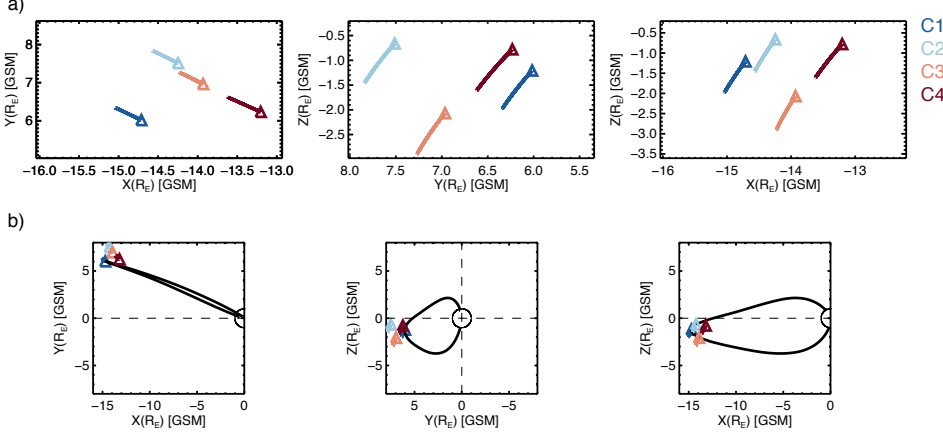


**Figure 1:** a) The locations of the Cluster spacecraft in the X-Y, Y-Z, and X-Z GSM planes, from
left to right, respectively, at 00:00 UT on 12 October 2006, marked by the triangles. The
trajectories from 00:00 UT to 00:55 UT are marked by the solid lines. The spacecraft are
color-coded according to the key on the right. b) As in a), with a zoomed-out view. The Earth
is shown by the solid circle. A TA15 model magnetic field line passing through the location of
C1 is shown as the solid black line.





The IMF measurements used in this study were provided by the OMNIweb database at 1-
minute resolution, having been first propagated from L1 to the bow shock nose (King &
Papitashvili, 2005).

*2.2. SuperDARN Data*
The ionospheric observations presented in section 3.3 were provided by the Super Dual
Auroral Radar Network (SuperDARN), an international collaboration of 36 ground-based
radars (Nishitani et al., 2019) that make line-of-sight Doppler measurements of the
horizontal motion of the ionospheric plasma every few seconds (e.g. Chisham et al., 2007).
Here, we use 2-min ionospheric convection maps created by fitting the line-of-sight $\mathbf{E} \times \mathbf{B}$
velocity data to an eighth order expansion of the ionospheric electric potential in spherical
harmonics using the technique of Ruohoniemi & Baker (1998), implemented in the Radar
Software Toolkit (RST version 4.2, 2018). To accommodate intervals with limited data
availability, the data are supplemented with values derived from a statistical model
parameterized by IMF conditions. This is a well-established technique that has been
thoroughly discussed by, e.g., Chisham et al. (2007). The convection maps we present
employ the commonly used model of Ruohoniemi & Greenwald (1996). As a check on the
sensitivity of the maps to the choice of model input, we also tested the fitting using the
alternative model of Thomas and Shepherd (2018) and found that this has little impact on
the maps and no impact on our conclusions.

As a further measure to ensure that the choice of model is not critical to our results, we
chose not to use the concurrent IMF vector to parameterise the background model. In this
case, because we are using the SuperDARN data to provide evidence in support of the
expected large-scale influence of IMF $B_y$, we deemed it inappropriate to include model data
already parametrised by IMF $B_y$. We instead specify a nominal southward IMF with zero $B_y$
component in our analysis, to ensure that a background model with no pre-existing IMF $B_y$
influence is used. Although this might result in the patterns we show being less accurate
overall, especially in regions of poor data coverage, it will ensure that any $B_y$-associated
asymmetry in the maps is driven by the radar data from our interval of study, and not the
background model. This is discussed further in section 4.1, below.









**3 Observations**

In this section we present observations of the IMF, magnetotail magnetic field and plasma
flow, and ionospheric convection from an interval on 12 October 2006.

*3.1 IMF Observations*
Figure 2 presents an overview of the spacecraft data from an extended interval around our
period of specific interest for broader context. In Figure 2a, we show a time-series of the
IMF $B_y$ and IMF $B_z$ data from 20:00 UT on 11 October to 01:00 UT on 12 October 2006. These
data reveal that IMF $B_y$ was generally positive for several hours prior to the fast flow
interval, with IMF $B_z$ predominantly negative. There were three small intervals of negative
IMF $B_y$ at ~ 21:35 UT, 23:00 UT and 23:40 UT and we discuss the possible ramifications of
these, and our treatment of them, in section 4.1.

*3.2 Cluster Spacecraft Observations*
In Figure 2b, we present the in-situ magnetic field and plasma measurements from the
Cluster spacecraft across the interval 00:00 – 00:55 UT.

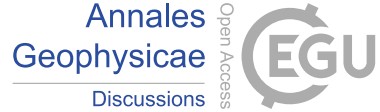



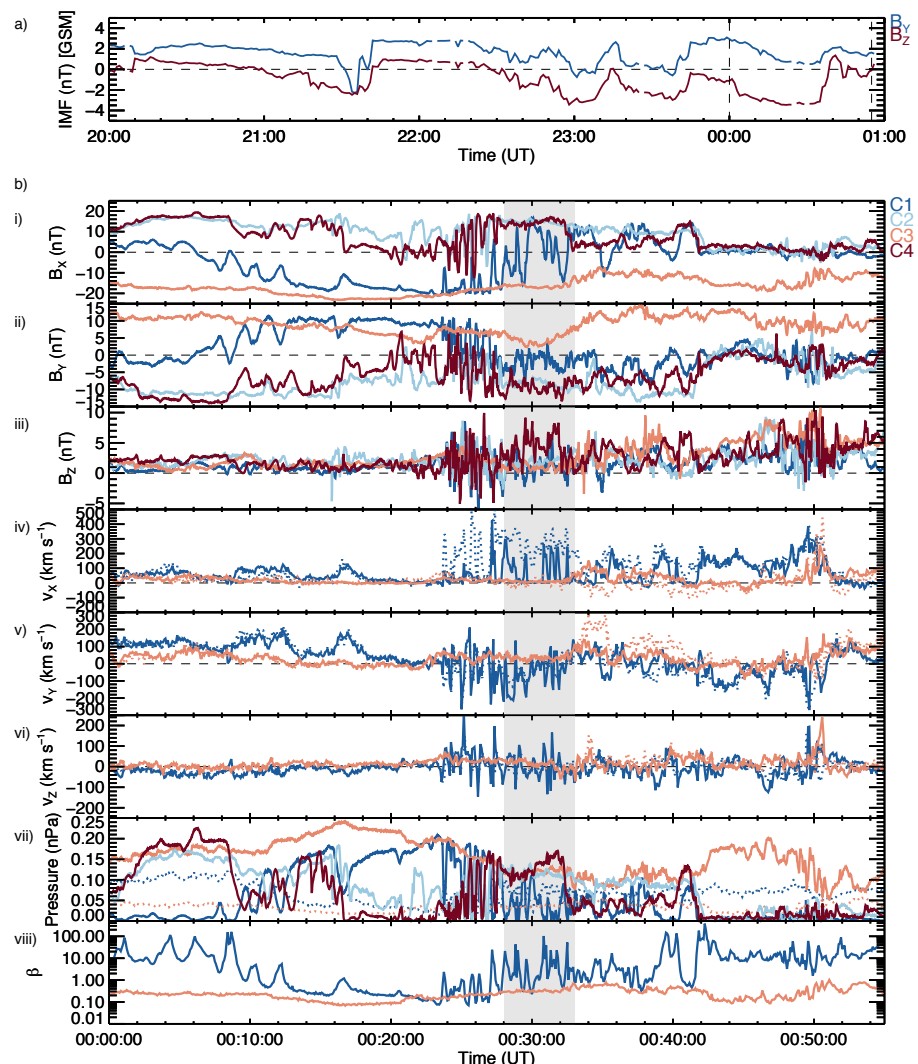


**Figure 2:** a) A plot of the IMF time series data for the IMF $B_y$ (blue) and IMF $B_z$ (red)

components, from 20:00 UT on 11 October 2006 to 01:00 UT on 12 October 2006. The

vertical dashed lines indicate the start (00:00 UT) and end (00:55 UT) of the interval of

Cluster data (below). b) The in-situ Cluster spacecraft measurements. Shown first is the local

magnetic field data, i) $B_x$, ii) $B_y$ and iii) $B_z$, followed by the bulk ion velocity data, iv) $v_x$, v) $v_y$,

and vi) $v_z$ (dotted lines). The field-perpendicular component of the ion flow (indicative of

the **E x B** convection) is shown in panels iv) to vi) by the solid lines. In panel vii) the magnetic

$\left(\frac{B^2}{2\mu_0}\right)$ and thermal ion ($nkT$) pressures are shown by the solid and dotted lines respectively,



and in panel viii) the ion plasma beta from C1 and C3 is shown. All data are labelled
according to the color-coded key on the right-hand side. The time-interval between the gray
shaded region marks our specific interval of interest (discussed in text).


At ~00:06 UT, C1 crossed from the northern hemisphere into the southern hemisphere,
illustrated by the sign change in $B_x$ from positive to negative shown in Figure 2b i).
Coincident with this, the observed $B_y$, shown in Figure 2b ii) turned from negative to
positive. Figure 2b iv) reveals that up until ~00:24 UT, the earthward flow measured by both
C1 and C3 was generally low in magnitude ($v_x < 100$ km s$^{-1}$). The $v_y$ component of the flow,
shown in Figure 2b v), remained steadily duskward ($v_y > 0$) at C1 and duskward or close to
zero at C3. The $v_z$ component of the flow in Figure 2b vi), measured by C1 and C3 was
effectively zero. During this period, the Cluster spacecraft that resided in the northern
hemisphere (predominantly C2 and C4), observed $B_y < 0$, and the spacecraft which resided
in the southern hemisphere (predominantly C1 and C3) observed $B_y > 0$. Occasionally a
spacecraft encountered the current sheet ($B_x = 0$) at which point it observed $B_y = 0$. We
comment on the significance of these magnetic field observations in section 4.2.

After ~00:24 UT, C1 began to observe a period of enhanced earthward flow
($v_x > 300$ km s$^{-1}$) and variable dusk-dawn flow, concurrent with sudden variation in the local
$B_x$ component. Similarly, C2 and C4, but not C3, observed large magnitude (> 20 nT) rapid
variations in $B_x$, which appear to have an apparent timescale of around a minute and which
we attribute to a flapping of the current sheet. As well as rapid variations in $B_x$, both the $B_y$
and $B_z$ components of C1, C2 and C4 seemed highly variable. As perhaps to be expected,
these variations in the magnetic field were accompanied by significant variations in the
magnetic pressure of ~0.15 nPa, as shown by the solid lines in Figure 2b vii).

Unlike the other spacecraft, C3 remained in the southern hemisphere throughout the entire
interval and did not observe the rapid fluctuations in $B_x$. Between 00:28 – 00:33 UT (the gray
shaded region), C1 began to repeatedly and rapidly cross the current sheet, as previously
experienced by C2 and C4, whilst continually observing enhanced earthward flow and





variable dusk-dawn convective flow ($v_{\perp y}$). Across the entire interval, the plasma beta, $\beta$,
indicated in Figure 2b viii), measured by C3 remained above ~0.1, with C1's measured $\beta$
ranging from 0.1 to over 100. This is consistent with the fact that C1 was continually
crossing the current sheet at the center of the plasma sheet, where $\beta$ is larger (Baumjohann
et al., 1989). It is this interval of current sheet crossing and variable flow observed by C1
that we focus on below and is presented in more detail in Figure 3.

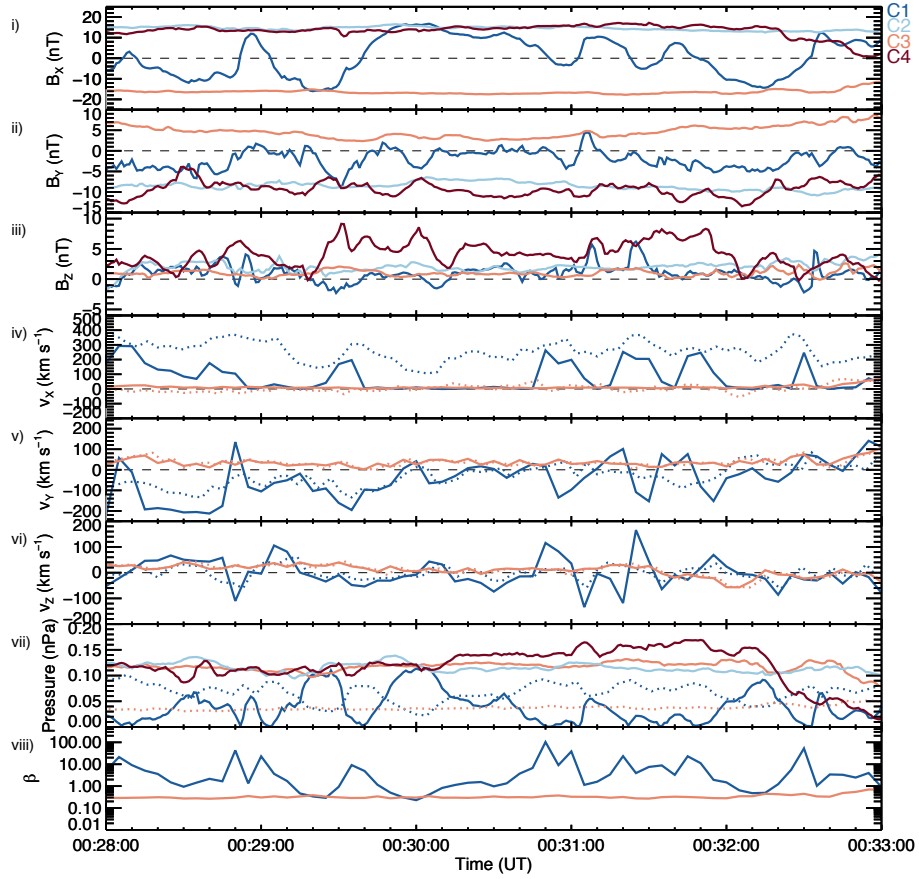


**Figure 3:** As in Figure 2b, but for the interval 00:28 – 00:33 UT on 12 October 2006.

Figure 3 i) conveys the extent of the large-amplitude $B_x$ variations observed by C1 between
00:28 and 00:33 UT. $B_x$ was generally fluctuating between positive and negative values
throughout the five-minute interval, with a minimum at ~ −16 nT and maximum at ~17 nT.
The magnetic pressure at C1 shown by the solid blue line in Figure 3 vii) is consistent with



the idea that C1 was crossing the current sheet, as this generally reached minima at the
center of each current sheet crossing ($B_x \approx 0$). The $B_y$ component (Fig. 3ii) measured by C1
generally remained negative and highly variable for the entire interval, with a number of
large negative enhancements and a few small positive excursions. It is particularly of note
that when C1 was below the neutral sheet, as implied by a negative $B_x$ component, $B_y$ was
almost always negative. As we discuss in section 4.2, this is inconsistent with what we would
expect based on the location of the spacecraft and also inconsistent with any expectation
that a positive IMF $B_y$ should have penetrated into the tail. The $B_z$ component (Fig. 3iii)
generally remained positive with some small negative excursions.

Unlike C1, C2-4 measured generally steady $B_x$ throughout this five-minute period. C2 and C4
measured positive $B_x$, indicating that they were above the neutral sheet, and C3 measured
negative $B_x$, indicating that it was below the neutral sheet. Similarly, $B_y$ was steadily negative
for C2 and C4 and steadily positive for C3. Again, we note the inconsistency between the C1
and C3 observations of $B_y$; when in the southern hemisphere C1 generally observed $B_y < 0$,
whereas C3 observed $B_y > 0$. On a few separate occasions C1 did briefly observe $B_y > 0$ (e.g.
at 00:31:05 UT) but at these times C1 was located above the neutral sheet ($B_x > 0$), while C2
and C4 observed $B_y < 0$ above the neutral sheet. These variations in $B_y$ imply the observation
of a 'kink' in the field at the location of C1, the ramifications of which are discussed further
in section 4.2.

At times when $B_x$ observed by C1 was negative, indicating that C1 was below the neutral
sheet, C1 generally observed negative (dawnward) $v_{\perp y}$ (Fig. 3v) with a magnitude varying
between 100 and 200 km s$^{-1}$. At times when $B_x$ became positive, indicating that C1 was
above the neutral sheet, C1 tended to observe positive (duskward) $v_{\perp y}$, although this flow
barely reached 100 km s$^{-1}$. The enhancements in $v_{\perp y}$ (both positive and negative) were
generally accompanied by negative enhancements in $B_y$. Across the interval, there was a
near continual $v_x > 200$ km s$^{-1}$ flow (blue dotted line in Fig. 3iv), peaking at almost 400 km
s$^{-1}$, with concurrent peaks in the convective $v_{\perp x}$ component (solid blue line) of at least
200 km s$^{-1}$. The convective flow measured by C3, however, was generally very weak ($|v_\perp| <$
50 km s$^{-1}$) throughout this period (solid orange line in Fig 3iv). $v_z$ (Fig. 3vi), as measured by





both C1 and C3 remained low in magnitude (< 100 km s$^{-1}$) for the duration of the interval,
with a few $v_{\perp z}$ excursions above 100 km s$^{-1}$ observed by C1. The most significant
enhancements in $v_{\perp z}$ seen by C1 appeared to occur in conjunction with the rapid current
sheet crossings between 00:30:50 and 00:32:00 UT. We discuss the implications of these
observations in the context of the upstream IMF conditions and large-scale magnetospheric
morphology in section 4.


*3.3 Ionospheric Convection Observations*

To provide the large-scale context in which we can interpret the more localized
observations from the Cluster spacecraft we show ionospheric convection observations in
Figure 4. In Figure 4a we present a series of four 2-minute integration SuperDARN maps of
the northern hemisphere ionospheric convection pattern, beginning at 00:24 UT, and
ending at 00:34 UT, which encompasses our specific interval. In all maps, plasma is flowing
anti-sunward across the polar cap at high latitudes, also with a strong duskward sense, with
the direction of the convection reversing in the pre-midnight sector before returning
sunward at lower latitudes.

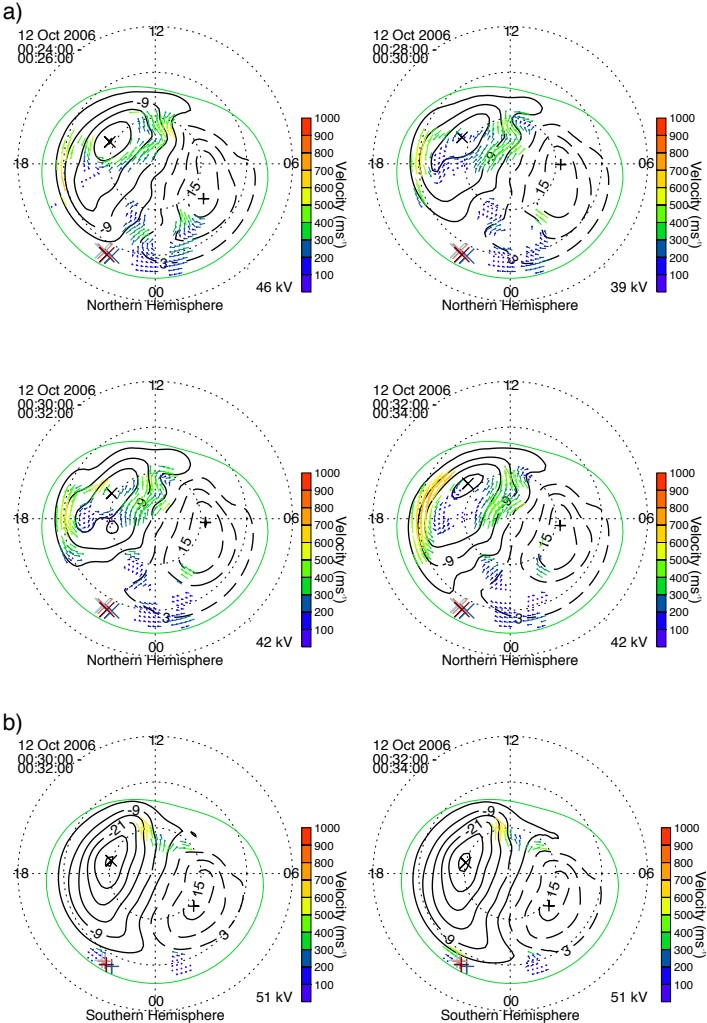


**Figure 4:** Maps of the ionospheric plasma convection derived from SuperDARN

observations. Midnight is to the bottom of each map, noon to the top, dusk to the left and

dawn to the right. The solid and dashed black lines represent the plasma streamlines and

are the contours of the electrostatic potential. Flow vectors are plotted at the locations of

radar observations and these are color-coded based on the magnitude of their velocity. a)

Four 2-minute northern hemisphere maps from 00:24 – 00:26, 00:28 – 00:30, 00:30 – 00:32

and 00:32 – 00:34 UT, respectively. b) Two 2-minute southern hemisphere maps from 00:30

– 00:32 and 00:32 – 00:34 UT, respectively. On each northern (southern) hemisphere map,

the footpoints of the Cluster spacecraft constellation are shown by the X's (+'s), mapped
using the TA15 model.


Owing to the coupled nature of the magnetosphere-ionosphere system, the observed
ionospheric convection pattern is indicative of the global-scale magnetospheric convection
(Cowley, 1981). In this case, the typical symmetrical twin-cell convection pattern has been
rotated clockwise, with the dawn cell extending across into the pre-midnight sector,
indicative of convection that has been driven under the influence of a positive IMF $B_y$
component (e.g. Reistad et al., 2016, 2018). On each northern hemisphere map, the
footpoints of the Cluster spacecraft constellation are indicated by the crosses (X), mapped
using the TA15 model with the same parameterisation described in section 2.

Figure 4b shows two 2-minute integration SuperDARN maps of the southern hemisphere
ionospheric convection pattern, beginning at 00:30 UT, and ending at 00:34 UT. The
associated footpoints of the Cluster spacecraft are indicated by the plus signs (+). Although
the coverage of radar data is much less than in the northern hemisphere, there are data in
the pre- and post-midnight sectors which appears to be influencing the location of the flow
reversal region at the nightside end of the dusk cell. Opposite to the northern hemisphere
case, it is the dusk cell in the south which is extending towards, or just beyond, the midnight
meridian. This is also consistent with a large-scale positive IMF $B_y$ influence, owing to the
expected north-south asymmetry of the influence of IMF $B_y$ in the magnetosphere (e.g.
Pettigrew et al., 2010). The significance of these observations is further discussed in section

336   4.1.


**4. Analysis and Discussion**

We have presented observations of a dynamic interval of plasma flows and magnetic field in
the Earth's magnetotail. In this section we discuss our rationale for interpreting the flows as
being inconsistent with large-scale magnetotail untwisting and our interpretation of their
relationship to current sheet flapping.





*4.1 Evidence for an inconsistency with large-scale magnetotail untwisting*
During the five-minute interval studied (00:28 – 00:33 UT) C1 measured a continually
fluctuating $B_x$ component (Fig. 3i), indicative of multiple crossings of the tail current sheet.
C1 was the only spacecraft to measure this signature across the interval (although similar
signatures had been observed a few minutes earlier by C2 and C4). C1 also measured a
series of earthward convective magnetotail fast flows with varying dusk-dawn components.
The data in Figure 3 i) and Figure 3 v) illustrate that when $B_x$ was positive (negative), a
duskward (dawnward) $v_{\perp y}$ was generally observed. Additionally, the data in Figure 3 ii)
show that C1 tended to observe a negative $B_y$ component. According to the magnetotail
untwisting hypothesis (e.g. Pitkänen et al., 2015), these flow and magnetic field
observations are consistent with a negative IMF $B_y$ penetration. The IMF data presented in
Figure 2a, on the other hand, revealed that IMF $B_y$ was generally positive for several hours
prior to the fast flow interval (00:28 – 00:33 UT). Based on the IMF data alone, therefore,
one might expect that a positive IMF $B_y$ will have penetrated into the magnetosphere and
thus ought to have determined the "expected" dusk-dawn direction of the flow. In that
case, the flows observed here would have a dusk-dawn sense that is not explained by
current theoretical models of magnetotail untwisting (e.g. Grocott et al., 2007).  There are a
number of possible explanations for this discrepancy and we address each one in turn.

The first possibility is that our conclusion regarding what is the expected dusk-dawn
asymmetry is incorrect. We noted in section 3.1 that there were three small negative IMF $B_y$
excursions prior to our Cluster observations interval. Although the propagation of the IMF to
the bow shock is accounted for in the OMNI data, there is uncertainty regarding the time it
takes for the IMF $B_y$ to 'propagate' into the magnetotail. Uncertainties in IMF $B_y$ propagation
times (e.g. Case & Wild, 2012) have previously been cited as an explanation for observing an
unexpected asymmetry (e.g. Pitkänen et al., 2013). Studies such as Tenfjord et al. (2015,
2017) and Case et al. (2018), for example, have suggested a reconfiguration time (to the
prevailing IMF $B_y$ conditions) for nightside closed field lines of around 40 minutes. At ~00:28
UT (the beginning of our specific interval of interest), the IMF $B_y$ had been positive for
around 50 minutes. Based on the Tenfjord timescale, this would thus imply that our interval
was wholly IMF $B_y > 0$ driven. Other studies, on the other hand, such as Browett et al.
(2017), have shown that longer timescales of a few hours may be important.






However, for such long timescales to play a role one would expect to have observed a
relatively persistent IMF $B_y$ component during that time. The integrated IMF $B_y$ over the
hours prior to our interval was certainly convincingly $B_y$-positive, and it seems highly unlikely
that a few minute-long fluctuations into the opposite IMF $B_y$ polarity, 1 or 2 hours prior to
the flows we observed, could have a significant influence. We can thus be confident that
positive IMF $B_y$ was governing the global magnetospheric dynamics in this case.

Despite this convincing argument that the IMF data alone imply a positive IMF $B_y$
penetration, we performed an additional analysis to further ensure that these negative
excursions did not lead to a change in the global nature of the magnetosphere-ionosphere
system. We inspected the concurrent northern hemisphere SuperDARN data (presented in
Figure 4a) to provide evidence of the large-scale convection pattern. If the large-scale flow is
consistent with a positive IMF $B_y$ component, then the magnetotail flows that we observed
must be deviating from this for some reason. The SuperDARN data indeed confirm that the
large-scale morphology of the system was consistent with a positive IMF $B_y$ component (e.g.
Lockwood 1993; Grocott et al., 2017; Reistad et al., 2018). This can be inferred from the
general shape of the convection pattern, whereby across multiple maps (00:24 – 00:34 UT)
the pattern was rotated clockwise, with the dawn cell having extended into the pre-
midnight sector. That this is the expected convection pattern for an IMF $B_y$-driven
magnetosphere is also supported by the concurrent low level of geomagnetic activity. The
auroral AU and AL indices (not shown) confirm that this interval is geomagnetically quiet
(AU and |AL| both less than (or of the order of) 10 nT), such that the nightside ionospheric
convection asymmetry should be driven by IMF $B_y$ rather than conductivity-driven features
such as the Harang discontinuity which might otherwise complicate the auroral zone flows
(e.g. Grocott et al., 2007; Grocott et al., 2008; Reistad et al., 2018).

The validity of the convection observations is further supported by the coverage of nightside
data which were used to constrain the model convection pattern. The data used to create a
SuperDARN convection map are supplemented by data from a statistical model (in this case
Ruohoniemi & Greenwald, 1996) which is typically parameterised by the instantaneous IMF
conditions. In the case that there is a lack of real data coverage, a created SuperDARN map





will be strongly influenced by the model data, as opposed to real data, and thus would
reflect a prediction of convection based on the IMF conditions. The maps shown in Figure 4a
illustrate that there were dozens of SuperDARN vectors in the midnight sector which were
fitted to create the global convection maps. To confirm that these data were sufficient, and
that the observed large-scale convection pattern was not being driven by model data, we
parameterised the model in our analysis with IMF $B_y$ = 0. Despite this, a clear IMF $B_y$-
asymmetry exists, thus demonstrating that the observed large-scale IMF $B_y$ > 0 global
convection patterns must be data-driven.

A second possible explanation for the discrepancy between the dusk-dawn direction of the
local and global-scale convection concerns the certainty with which we can determine the
location of the spacecraft with respect to the large-scale convection pattern. The untwisting
hypothesis relies on the assumption that the convection cell to which the spacecraft is
connected should be a factor of only hemisphere and the sense of IMF $B_y$ (e.g. Pitkänen et
al., 2013, 2017). In other words, as discussed above, for $B_y$ > 0, the hypothesis dictates that
C1 ought to be located on the dawn cell when above the neutral sheet and the dusk cell
when below, at least in the case that the spacecraft is close to midnight (Grocott et al.,
2007). This might be true statistically, but it is not clear how valid an assumption it might be
when trying to interpret observations from a single event in the presence of a highly
dynamic neutral sheet. It also fails to account for the dusk-dawn location of the spacecraft,
which in this case was $6 \lesssim Y \lesssim 7$ R$_E$. If, as a result, the spacecraft was actually located on the
dusk cell when above the neutral sheet, and on the dawn cell when below the neutral sheet,
then the sense of the observed plasma sheet flows would actually be consistent with the
large-scale convection.

One way to specify which cell the spacecraft is located within is to map its location into the
ionosphere. This has been done using TA15 and is shown by the crosses (X) on the northern
hemisphere convection maps and by plus signs (+) on the southern hemisphere convection
maps, in Figures 4a and 4b, respectively. This mapping suggests that the assumption above
is correct.





Consider first the northern hemisphere map from 00:30 – 00:32 UT in Figure 4a: the
spacecraft map close to the dawn cell, such that the duskward flow that C1 observed there
would seem to be inconsistent. However, it is worth considering that the pre-midnight
location of the spacecraft, the proximity of the mapped footpoints to the dusk cell, and the
level of uncertainty generally accepted to be present in field line mapping, may give
credence to the possibility that the spacecraft actually mapped to the dusk cell in the
northern hemisphere. If this was the case, then the northern hemisphere flows observed by
C1 would actually be consistent with the large-scale convection pattern. However, if we
consider the southern hemisphere maps in Figure 4b we can be more certain of which cell
the spacecraft map to. Owing to the IMF $B_y$ positive nature of the convection (i.e. the more
extended southern hemisphere dusk cell) and the pre-midnight location of the spacecraft,
the footpoints are located quite convincingly on the dusk cell. This is despite the dusk-dawn
asymmetry being less pronounced than that seen in the northern hemisphere (and the
associated poorer coverage of southern hemisphere SuperDARN data). When below the
neutral sheet C1 observed dawnward flows, meaning it would have to have been on the
southern hemisphere dawn cell to be consistent with the large-scale convection, which is
clearly not the case. It seems much more likely, therefore, that C1 observed flow that was
associated with localized magnetic field dynamics rather than being a signature of the large-
scale convection.

*4.2 Evidence for a local perturbation in the magnetotail*
The lack of consistency with the large-scale convection leads us to a third explanation for
our observations, which is that there is a local perturbation within the tail that is
independent of any large-scale, IMF $B_y$-controlled asymmetry associated with magnetotail
untwisting. This is supported by the observations from the other Cluster spacecraft. The
low-level of flow seen by C3 is mostly duskward (Fig. 3v) and therefore consistent with the
idea of untwisting under IMF $B_y > 0$, given its southern hemisphere location. Further, in
Figure 2b v), up until the rapid $B_x$ variations began at ~00:24 UT, fast duskward flow in the
southern hemisphere was also seen by C1. The fact that C3 continued to then observe
steady duskward flow, and no significant $B_x$ change, suggests that the change in the nature
of the C1 observations after 00:24 UT must in-fact be due to some localized process that





was responsible for driving the dawnward component of the flows which was only observed
by C1.

This idea of a local perturbation is also supported by the variations in the local $B_y$
component. Figure 3 ii) illustrates the in-situ variations in $B_y$ with time across the interval.
Despite there clearly being positive IMF $B_y$ penetration globally (as confirmed by inspection
of the OMNI and SuperDARN data), C1, C2 and C4 all recorded mostly negative local $B_y$
values. In the studies of, e.g., Pitkänen et al. (2013, 2017) this observation would have been
offered as evidence of a negative of IMF $B_y$ penetration, thus supporting the untwisting
hypothesis. However, it is important to note that a negative local $B_y$ component may be
wholly consistent with positive IMF $B_y$. There are, in fact, multiple sources of $B_y$ in the tail,
such as magnetotail flaring (Fairfield, 1979), as well as tilt effects and current sheet warping
(see e.g. Petrukovich et al., 2005), in addition to a penetration of the IMF $B_y$. To fully
interpret the magnetic field observations, we must therefore consider the possible effects
of these phenomena on the presence of $B_y$ in the tail at the specific location of each
spacecraft.

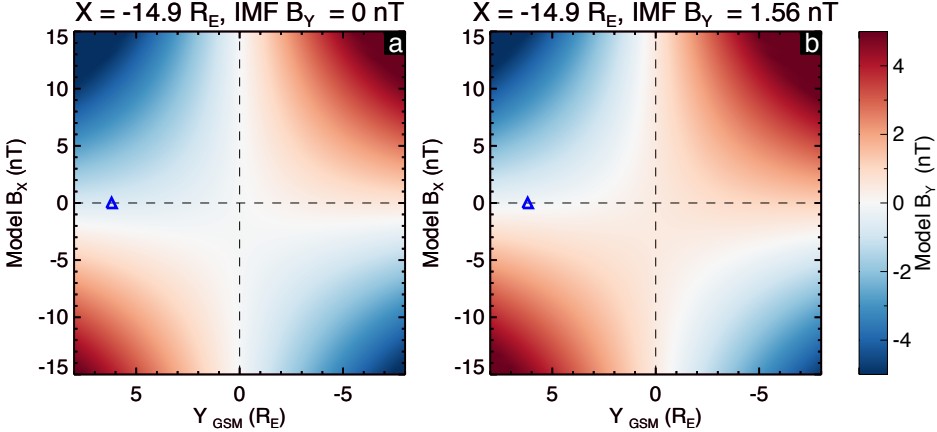


**Figure 5**: TA15 model magnetic field data. In each case, plotted is Y vs $B_x$ [GSM], (at
X=−14.9 $R_E$, i.e. the X position of C1 at ~00:28 UT on 12 Oct 2006), with the TA15 modelled
$B_y$ value shown by the color bar on the right. The blue triangle shows the Y-location of C1, at
$B_x = 0$. In panel (a) we have imposed IMF $B_y = 0$, and for panel (b) we have used the 1-hour
mean-averaged IMF $B_y$ (+1.56 nT) in the hour prior to 00:28 UT.




To aid in this interpretation, we present TA15 model magnetic field data in Figure 5, to
provide an indication of the expected background $B_y$-component at the time of our interval.
These data, from X = −14.9 $R_E$, are plotted against Y [GSM]-position on the horizontal axis,
and against the $B_x$-component on the vertical axis. We have reversed the conventional
direction of the horizontal axis (negative to positive from left to right) to be consistent with
a view looking earthward from downtail. In panel (a) we show the field for the case that IMF
$B_y$ = 0 and in panel (b) the case that IMF $B_y$ = +1.56 nT (the 1-hour mean-averaged IMF $B_y$ in
the hour prior to 00:28 UT). The first conclusion we can make from consideration of the $B_y$
component in Figure 5a is how, even under no IMF $B_y$ penetration, a 'background' $B_y$ value
will exist in the tail purely dependent on location. In such a 'symmetric' tail, one would
expect the background $B_y$ value to appear as one moves away from midnight toward the
dusk-dawn flanks, as well as further above and below the neutral sheet. Pre-midnight, we
would expect to observe negative $B_y$ above the neutral sheet ($B_x > 0$), and positive $B_y$ below
the neutral sheet ($B_x < 0$), with the opposite effect post-midnight. This is the effect known as
magnetotail flaring (Fairfield, 1979).

The data in Figure 5a also show the effect of the negative (tailward) dipole tilt (as
appropriate to our study interval) and current sheet warping on the local $B_y$ component.
According to Petrukovich (2011) the current sheet warping (controlled by the dipole tilt) is
expected to add a negative $B_y$ component pre-midnight and a positive $B_y$ component post-
midnight. Furthermore, the 'even tilt' effect is expected to add a negative $B_y$ component to
both the pre and post-midnight sectors for a negative tilt. This leads to the effect seen in
Figure 5a where in the pre-midnight sector, the location of the $B_y$ polarity change occurs in
the southern hemisphere (at $B_x \approx -3$ nT).

Figure 5b illustrates the scenario relevant to our case study, where we have additionally a
global positive IMF $B_y$ penetration. This additional positive $B_y$ has the effect of moving the
location of the pre-midnight $B_y$ polarity change back up towards the neutral sheet. This
explains why the Cluster spacecraft observed $B_y \approx 0$ when $B_x \approx 0$ during the few tens of
minutes prior to our interval, as noted in section 3.2. This also explains why C2-3 and C4
observed the polarity of $B_y$ that they did throughout the interval. It is thus clear that positive



IMF $B_y$ penetration does not mean we should expect to observe positive $B_y$ everywhere in
the tail, rather, it simply means that there is expected to be some positive $B_y$ perturbation
to the already present 'background' $B_y$ at a particular location. As Figure 5b demonstrates,
C2 and C4 (located above the neutral sheet) are expected to have observed negative $B_y$
even though positive IMF $B_y$ has penetrated into the magnetotail. The background $B_y$
expected at their location (pre-midnight, $B_x > 0$), is negative and the IMF $B_y$ -associated
perturbation was not large enough to enforce a sign change in $B_y$.

The Cluster spacecraft in our study were all located pre-midnight (+Y GSM). From Figure 3,
C2 and C4 observed positive $B_x$, and negative $B_y$, and at ~00:28 UT were located at around
Z = −1 $R_E$ (Figure 1). C3, however, observed negative $B_x$ and positive $B_y$, and was located at
around Z = −2.5 $R_E$. The location of the neutral sheet crossing at ~00:28 UT can therefore be
said (locally) to have been somewhere between −1 and −2.5 $R_E$ in Z. C1 was located at
around Z = −1.5 $R_E$ and, throughout the five-minute interval, observed a $B_x$ which continually
fluctuated from positive to negative, yet observed mostly weakly negative $B_y$. For $B_y$ to have
remained negative, despite C1 moving above and below the neutral sheet, suggests that
there was a $B_y$ negative 'kink' in the magnetotail that was localized to the vicinity of C1. This
is further supported by the fact that numerous (albeit brief) positive $B_y$ excursions occurred
when C1 was above the neutral sheet (as noted in section 3.2). We use the term 'kink' to
highlight a deformation in the nearby field lines which results in the observed perturbations
to the local $B_y$ component. We suggest that this deformation could be relatively small in
terms of field line length, much like a kink in a cable or wire. In the following section, we
investigate this kink in relation to the observed current sheet flapping.


*4.3 Evidence for current sheet flapping as a source of the asymmetric flows*
If a localized magnetic field perturbation was associated with the lack of observation of the
expected dusk-dawn flow for magnetotail untwisting, investigating its cause seems a
worthwhile endeavour. The clear sinusoidal-like variation in $B_x$ observed by C1, which is
evidence of current sheet flapping (e.g. Runov et al., 2009), provides us with a starting point
for this investigation. This flapping must be highly localized as at the time of our five-minute
flow interval (00:28 -00:33 UT), only C1 observed the flapping. MVA analysis (Sonnerup &



Cahill, 1967) suggests that the flapping was a kink-like wave which was propagating
dawnward (Rong et al., 2015; Wu et al., 2016), and therefore may have been a source of the
observed dusk-dawn flow.

The causes of current sheet flapping have been discussed previously (Runov et al., 2009;
Wei et al., 2019). One such cause has been attributed to localized, periodical reconnection –
a process known to drive Bursty Bulk Flows (BBFs) in the magnetotail (Angelopoulos et al.,
1994; Zhang et al., 2016). In fact, BBFs excited directly as a result of reconnection in the tail
have been previously linked to magnetic fluctuations in the current sheet (Nakamura et al.,
2009; Wu et al., 2016). Examining the data presented in Figure 3 iii) and Figure 3 iv), we
note that C1 measured a generally positive $B_z$, with a few negative blips, as well as
continually fast ($v_x > 200$ km s$^{-1}$) earthward flow, peaking at over 370 km s$^{-1}$ with bursts of
enhanced convective flow ($v_{\perp x} > 200$ km s$^{-1}$) also apparent. These observations are fairly
consistent with (if slightly slower than) the original definition of a BBF (Angelopoulos et al.,
1994). This, along with the absence of similar flow observations in the C3 data, suggests that
C1 may have been located earthward of a localized reconnection site (owing to $B_z > 0$),
where persistent, localized reconnection was exciting fast earthward flow. The reconnection
process may then have been driving the current sheet flapping, inducing the localized kink in
the field, and ultimately controlling the dusk-dawn direction of the convective flow.


It is well known that the magnetic tension force is responsible for the acceleration of plasma
following reconnection (Karlsson et al., 2015). Our observations of a dusk-dawn flow
component may be related to the localized magnetic tension forces driving and directing
plasma flows in association with the flapping. In order to provide some scope to this
suggestion, we attempted to find the direction of the ***J*** **×** ***B*** forces acting on the plasma. We
used the curlometer technique (Dunlop et al., 1988, 2002), to estimate the average current
density, ***J***, flowing through the volume bound by the spacecraft tetrahedron. The ***J*** **×** ***B***
force density [N m$^{-3}$] is then calculated by taking the cross product of ***J*** with the average
magnetic field vector ***B*** from the four-spacecraft (Karlsson et al., 2015).





In order to check the validity of using the curlometer approach, we calculated the quality
parameter, $Q$, defined as $|\nabla \cdot \boldsymbol{B}|/|\nabla \times \boldsymbol{B}|$. It is generally accepted that a value of $Q < 0.5$ is
required for a current estimate to be valid. Hence, the value of $Q$, along with due
consideration of the spacecraft configuration and its orientation relative to the magnetic
field structure, may be used as a monitor of how reliable the curlometer approach is
(Dunlop et al., 2002). This is discussed further below, in reference to the analysis shown in
Figure 6.

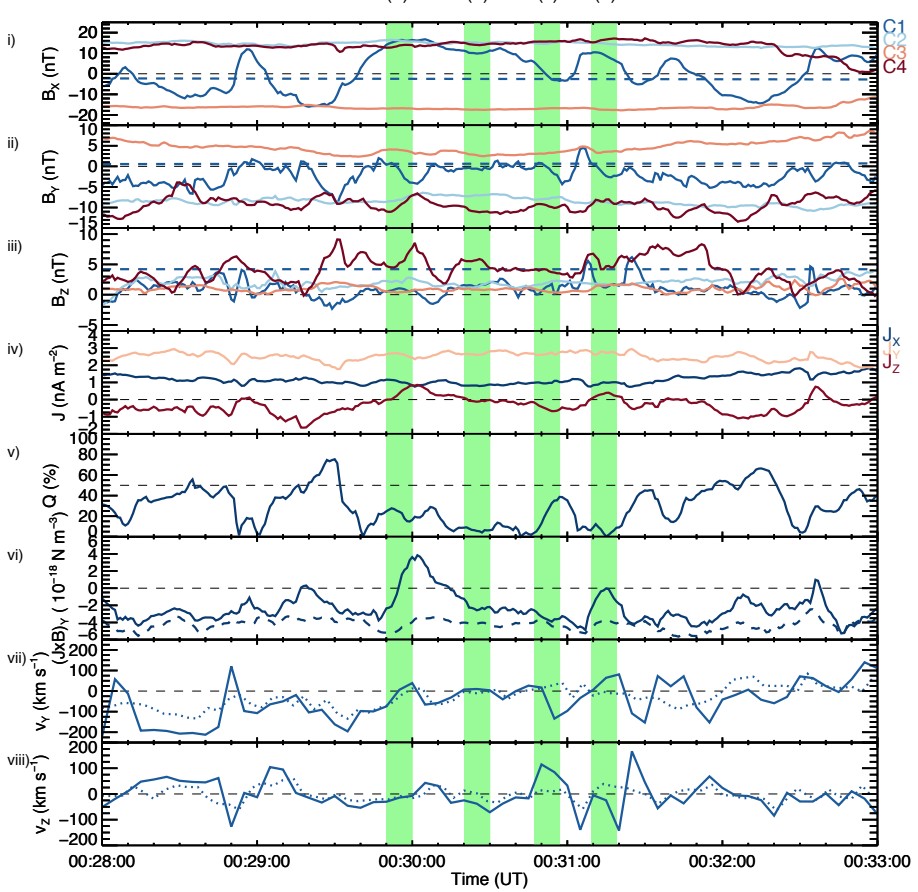


**Figure 6:** i-iii) The local magnetic field vector $\boldsymbol{B}$ ($B_x$, $B_y$, $B_z$) observed by C1-4, as shown
previously (solid lines) and the TA15 modelled $\boldsymbol{B}$ vector for C1 (dashed blue lines). iv) The
components of the current density vector $\boldsymbol{J}$ ($J_x$, $J_y$, $J_z$), v) $Q$, vi) ($\boldsymbol{J} \times \boldsymbol{B}$)$_Y$ (solid blue line) and ($\boldsymbol{J}$



× $\boldsymbol{B}$)$_y$ computed using the TA15 modelled C1 $\boldsymbol{B}$ (dashed line, discussed in-text), vii) $v_y$ ($v_{\perp y}$
in solid lines), observed by C1 and viii) $v_z$ ($v_{\perp z}$ in solid lines), also observed by C1. The green
highlighted regions labelled (a), (b), (c) and (d) correspond to four specific time-windows of
interest (discussed in-text).

Shown in Figure 6 i-iii) are the local magnetic field $B_x$, $B_y$ and $B_z$ components, as presented
previously. In Figure 6 iv) are the current density $J_x$, $J_y$ and $J_z$ components determined from
the curlometer analysis. In Figure 6 vi) is the dusk-dawn component of $\boldsymbol{J} \times \boldsymbol{B}$. In panels i-iii)
and vi), also shown is a dashed blue line. In panels (i-iii) this represents the TA15 modelled
magnetic field (see section 4.2) at the location of C1. In panel (vi) this represents the ($\boldsymbol{J} \times \boldsymbol{B}$)$_y$
force where $\boldsymbol{J}$ and the average $\boldsymbol{B}$ have been computed using the model field at the location
of C1 and the true magnetic fields measured by C2-C4, hereafter referred to as the 'model ($\boldsymbol{J}$
× $\boldsymbol{B}$)$_y$ force'. This has been computed to provide an illustration of what one would expect
the 'unperturbed' magnetic field of C1 and the associated ($\boldsymbol{J} \times \boldsymbol{B}$)$_y$ force to look like, in the
absence of any dynamical effects such as current sheet flapping or field line 'kinking'. Figure
6 v) suggests that our curlometer approach is generally appropriate, as $Q$ mostly remains
below 50% (horizontal dashed line) for the five-minute interval. We note that, unlike in
previous studies which have used the curlometer technique at inter-spacecraft separation
distances of << 1 $R_E$ (e.g. Dunlop et al., 2002; Runov et al., 2003), in our case the Cluster
spacecraft separation is large ($\gtrsim$ 1 $R_E$). Therefore, the curlometer is likely to be an
underestimate of the true current at these scale sizes. Critically, however, the spacecraft
configuration is such that the estimate of the direction of the currents should be stable.
Thus, although the volume enclosed by the spacecraft is greater than the scale sizes of the
current sheet flapping and kink, a reliable estimate of the direction of the net $\boldsymbol{J} \times \boldsymbol{B}$ force
within the enclosed volume may still be obtained.

Two key features of Figure 6 are apparent. Firstly, it appears as though the perturbations to
($\boldsymbol{J} \times \boldsymbol{B}$)$_y$, displayed in Figure 6 vi), are associated with the magnetic field perturbations
generally only observed by C1. Second, the dawnward flow bursts (reproduced in Fig. 6 vii)
tend to occur when ($\boldsymbol{J} \times \boldsymbol{B}$)$_y$ is more negative, with the weak duskward flow bursts occurring


when $(\mathbf{J} \times \mathbf{B})_y$ is less negative. We note that there is not a one-to-one correlation between
the $(\mathbf{J} \times \mathbf{B})_y$ and $v_{\perp y}$ data. This could well be due to the large volume over which $\mathbf{J} \times \mathbf{B}$ is
being averaged and we make no attempt to interpret the detailed variations in $(\mathbf{J} \times \mathbf{B})_y$
implied by these data. However, as this region of space will contain the localized flapping
and kink, the calculated $\mathbf{J} \times \mathbf{B}$ should be influenced by these dynamics and hence still
provide an indication of the forces acting within that region. The consistency between the
direction of $(\mathbf{J} \times \mathbf{B})_y$ and $v_{\perp y}$ therefore suggests that the $\mathbf{J} \times \mathbf{B}$ force associated with the
current sheet flapping is exerting some level of control over the direction of the convective
flow. We also note that the $(\mathbf{J} \times \mathbf{B})_y$ force is effectively always less negative than the model
$(\mathbf{J} \times \mathbf{B})_y$ force. As can be seen in Figure 6 vi), the model $(\mathbf{J} \times \mathbf{B})_y$ force is acting steadily
dawnward, consistent with the duskward location of the spacecraft and suggesting that the
curlometer analysis is simply picking up the $(\mathbf{J} \times \mathbf{B})_y$ force associated with the 'background
curvature' of the magnetic field. Thus, we suggest that the positive deviations of $(\mathbf{J} \times \mathbf{B})_y$
from the model $(\mathbf{J} \times \mathbf{B})_y$ force are due to the perturbations (flapping and kinking) observed
by C1.



*4.4 Visualization of the observed dynamics*
In an effort to visualize these plasma sheet dynamics, we show in Figure 7 a series of
sketches that attempt to associate the observed magnetic field perturbations with the
observed dusk-dawn convective flows. The panels correspond to the four time windows
indicated on Figure 6 by the highlighted regions labelled a-d. In each panel, we indicate the
approximate relative position of the 4 Cluster spacecraft in GSM coordinates, and the
appropriate sense of $B_y$ measured by each spacecraft is shown by the purple arrows at each
spacecraft location (the Z-component of the field was in fact generally small, and has been
exaggerated here for illustrative purposes). We also superimpose nominal plasma sheet
field lines (again with an exaggerated extent in Z) that display the sense of $B_y$ implied by the
TA15 data presented in Figure 5 (long blue curved arrows). The dashed lines represent the
location of the neutral sheet at the end of each time window. This is tilted slightly, as

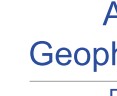
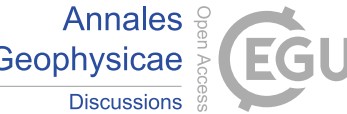

appropriate for IMF $B_y > 0$, but with the end-state of the "flap" of the current sheet implied
by the sign of $B_x$ observed by C1. In red is the perturbation to the field implied by the sign of
$B_y$ observed by C1.

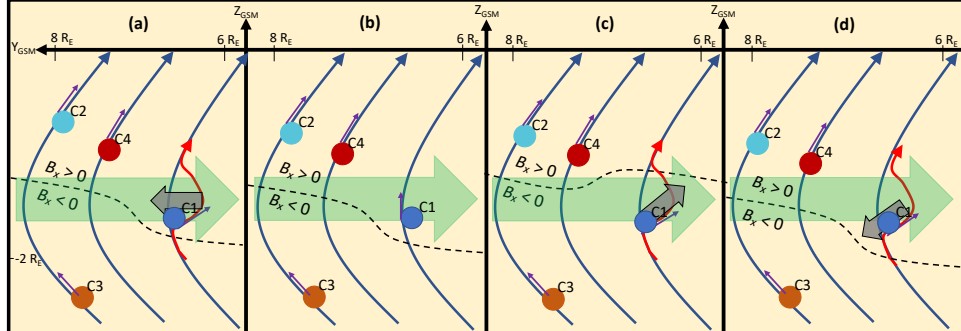


**Figure 7:** Schematic diagrams of the observed magnetic field perturbations and dusk-dawn
convective flows during the time-windows indicated in Fig. 6 by the highlighted regions. The
approximate locations of the four Cluster spacecraft relative to one-another in the Y-Z GSM
plane are indicated (not to scale) by the colored circles. The curved blue arrows represent
magnetic field lines, and the short purple arrow indicates the local sense of $B_y$ at the
location of each spacecraft. The dashed black line indicates the current sheet. In panels (a),
(b) and (d), the curved red arrow shows the 'kinked' magnetic field line. The long thick green
arrow shows the direction of the model $(\boldsymbol{J} \times \boldsymbol{B})_Y$ force associated with the background
curvature of the magnetic field, and the small thick gray arrow shows the direction of the
dusk-dawn convective flow observed by C1.


In Figure 7a C1 is located above the current sheet and measured negative $B_y$. A weakly
duskward convective flow was observed at this time (as indicated by the thick gray arrow),
consistent with the duskward sense of the $(\boldsymbol{J} \times \boldsymbol{B})_Y$ force, and opposite to the sense of the
model $(\boldsymbol{J} \times \boldsymbol{B})_Y$ force associated with the background curvature of the magnetic field. In
Figure 7b, C1 is still above the current sheet but measured $B_y \approx 0$ and no dusk-dawn
convective flow. In Figure 7c C1 is shown below the current sheet, where the background $B_y$
would be positive (see Fig 5b). C1 instead observed an increasingly negative $B_y$, which we





suggest is associated with the presence of the kink in the field. At the same time, C1 also
observed a convective plasma flow with dawnward and slightly upward (+Z) component
(thick gray arrow). We therefore suggest that the flow was associated with the
upward/dawnward flap of the current sheet, and that the dawnward sense of the flow likely
also resulted in the increase in negative $B_y$ seen during the time-window shown in Figure 6c.
In Figure 7d C1 is shown above the current sheet, where it observed a weakly negative $B_y$. In
this case, C1 observed a convective plasma flow with duskward and slightly downward (−Z)
component. Similarly to in Figure 7a, this flow occurred in concert with a positive
enhancement in $(\mathbf{J} \times \mathbf{B})_y$ relative to the model $(\mathbf{J} \times \mathbf{B})_y$. This flow would therefore seem to
be associated with the downward flap of the current sheet, and its duskward sense could
indicate that it is acting to reduce the negative kink in $B_y$ that is apparent over the time-
window shown in Figure 6d.

Whilst we acknowledge a degree of uncertainty in the details of the interpretation
presented above of the specific relationship between the flows and the field, it serves to
illustrate three observations about this interval of which we can be very certain: 1) The IMF,
ionospheric convection, and plasma sheet magnetic field observations all lead to the
expectation of an IMF $B_y > 0$ large-scale asymmetry in the magnetosphere. 2) The Cluster 1
spacecraft observed convective flow with a dusk-dawn component that was inconsistent
with current theories of IMF $B_y$-induced dusk-dawn flows associated with magnetotail
untwisting. We therefore note that the observations presented here cannot be attributed to
the current model of large-scale magnetotail untwisting.  3) Magnetic field perturbations
that were indicative of a localized current sheet flapping and dusk-dawn kink in the field
occurred coincident with the flows. It therefore seems likely that IMF $B_y$-driven asymmetries
are not the only mechanism by which a dusk-dawn component may be introduced into the
convective flow, with other dynamical processes also likely to contribute.









**5. Summary**

We have presented a case study from 12 October 2006 revealing a dynamic interval of
plasma flows and current sheet flapping, observed by the Cluster 1 spacecraft. The key
observations presented in this study may be summarised as follows:

• The OMNI data revealed that the IMF $B_y$ had been positive for several hours prior to
our interval of Cluster data, with the exception of three short-lived negative
excursions.
• The SuperDARN ionospheric convection observations revealed a large-scale
asymmetry consistent with IMF $B_y > 0$.
• C1 observed a changing $B_x$ magnetic field component, and associated duskward ($v_{\perp y}$
> 0) flow when in the northern magnetic hemisphere, and dawnward ($v_{\perp y} < 0$) flow
in the southern magnetic hemisphere.

Contrary to the results of a number of previous studies in the literature, during this
particular interval, the dusk-dawn sense of the convective magnetotail flows ($v_{\perp y}$) does not
agree with expectations based on the theoretical understanding of global magnetotail
untwisting and the prevailing positive IMF $B_y$ conditions. We instead attribute the flows to a
localized magnetic field perturbation, or 'kink' in the magnetotail, which appeared to be
independent of any large-scale IMF $B_y$ controlled asymmetry and may have been related to
the observed current sheet flapping. We attributed the current sheet flapping to being
driven by localized reconnection, itself inferred from the presence of the observed bursty
fast earthward flow ($v_{\perp x} \approx 200$ km s$^{-1}$). Analysis using the curlometer technique suggests
that the $\boldsymbol{J} \times \boldsymbol{B}$ force associated with the current sheet flapping could have been exerting a
level of control over the convective flow responsible for introducing the observed dusk-
dawn component.

Whilst it is known that variable dusk-dawn flow can occur in conjunction with current sheet
flapping, this case study has provided direct evidence that flapping can locally override the
expected IMF $B_y$ control of dusk-dawn magnetotail flow, in spite of clear global penetration





of IMF $B_y > 0$; consequently, resulting in the production of localized flows that do not agree
with the expected direction for global magnetotail untwisting. Further studies by the
authors are currently underway to determine if this is a frequent occurrence, and to
consider, and account for, localized tail dynamics more fully in a statistical analysis of the
magnetotail flows.

**Acknowledgements**

The authors would like to thank the FGM and CIS teams as part of the Cluster mission and
acknowledge the Cluster Science Archive (Laakso et al., 2010) as the source of the Cluster
data. We also wish to thank the OMNIWeb as the source of the solar wind and IMF data.
The authors acknowledge the use of SuperDARN data. SuperDARN is a collection of radars
funded by national scientific funding agencies of Australia, Canada, China, France, Japan,
South Africa, United Kingdom, and United States of America, and we thank the international
PI team for providing the data. The authors acknowledge access to the SuperDARN database
via BAS data mirror (http://bslsuperdarnc.nerc-bas.ac.uk:8093/docs/) and are grateful for
use of the Radar Software Toolkit (RST v4.2
https://zenodo.org/record/1403226#.Xy0u7y3MxTY) with which the raw radar data were
processed. We acknowledge the WDC for Geomagnetism, Kyoto, for use of the auroral
electrojet indices, which may be obtained from http://wdc.kugi.kyoto-u.ac.jp/aedir/. We are
also grateful to Haje Korth for providing the IDL Geopack DLM containing the Tsyganenko
magnetic field model routines and coordinate system conversions and wish to thank Nikolai
Tsyganenko for useful discussion of his magnetic field models. Finally, we are thankful for
the advice of Malcolm Dunlop regarding the applicability of the curlometer technique at
large spacecraft separations. This research was undertaken with the support of funding
from the following sources: Lancaster University Faculty of Science and Technology
studentship (JHL), STFC Consolidated grant no. ST/R000816/1 (NAC, AG), NERC standard
grant nos. NE/P001556/1 and NE/T000937/1 (MTW, AG).

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
