# Peer review of "Dynamics of Variable Dusk-Dawn Flow Associated with Magnetotail 1 Current Sheet Flapping 2 3 4 James H. Lane1, Adrian Grocott1, Nathan A. Case1, Maria-Theresia Walach1 5 6 1 Department of Physics, Lancaster University, Lancaster, UK"

_Annales Geophysicae, 2021_

## Author Response (AR1)

We would again like to thank all of the reviewers for their careful consideration of the manuscript and their valuable comments. Below, we have created a summary list which describes all of the **key** changes we have made to the manuscript as a result of the reviewer comments.

Below this list, for convenience we again show the reviewer comments and our responses during the open review process.

**Summary of Key Changes Made to Manuscript**

Please note that the line numbers quoted here refer to the 'tracked changes' manuscript.

- We have tweaked our argument that the 'flapping was overriding the IMF $B_y > 0$ control of the flow', and now suggest that the 'IMF $B_y > 0$ penetration at the Cluster location was unable to override (or overcome) the variable dusk-dawn flow associated with the flapping'. This has been made clear in the abstract (lines 26-28), introduction (lines 127-129), and summary (lines 940-942) in the revised manuscript.

- We have now introduced the concept of magnetotail flaring earlier in the manuscript, stating that this is important at Y-locations such as those during our event study, and that this can have effects on the dusk-dawn convection. We have also emphasised that the spacecraft observations are consistent with flaring. We have also highlighted the fact that previous studies (Pitkänen et al., 2013) have investigated IMF $B_y$-effects on the convection up to 7 $R_E$ towards the dusk-dawn flanks, why is why we feel it is important that we consider the possibility of this effect in our study – especially due to the fact that the observed flow is consistent with tail untwisting for an IMF $B_y < 0$ situation. We have drawn increased attention to the fact that it is really the observed dawnward flow in the pre-midnight, southern hemisphere, which was not consistent with being IMF $B_y$-controlled in any way. The duskward flow, meanwhile, could still be consistent with the larger-scale convection (even in the absence of asymmetry). See, for example, lines 79-91, 535-539, 555-558, 628-629, 914-918 in the revised manuscript.

- We have now highlighted the fact that the Cluster magnetic field observations do show a penetrated IMF $B_y > 0$, **when compared** to the TA15 model results (lines 872-873)

- As per R2, we have included both calculations for the $J \times B$ force, using both an average $B$ from all 4 Cluster spacecraft, and just the measured $B$ by C1, and tweaked our interpretation as appropriate. See lines 697-699, 755-783, 857-861, 922-935.

- As per R3, we have tidied up the axis-labels and changed to using the standard colour traces for the Cluster spacecraft.

- A number of minor tweaks, in relation to e.g. description of Figures. Please see the specific comments and responses, below.

**Response to Reviewers**

Please note that any line numbers quoted below refer to the original manuscript.

**Reviewer 1**

**Comment 1:** *First, the case locates at the dusk side of magnetotail where magnetic fields have strong positive/negative Y component. So that is not good place to estimate the little IMF By effect on the process at the magnetotail*

**Response 1:** We agree with the reviewer that the effect of magnetotail flaring towards the dusk-dawn flanks is apparent (Fig. 5 in the manuscript, and the C2-C4 data during the flapping interval). We haven't purposely chosen this location to try and estimate any IMF $B_y$ effects. We are investigating an interval of current sheet flapping (occurring at this location) and have considered that there *might* be an IMF $B_y$ effect. Previous studies such as Pitkänen et al. (2013, 2017) have investigated IMF $B_y$ control of magnetotail flows at up to ~ 7 $R_E$ towards the dusk-dawn flanks which revealed a clear dependence of the flows on IMF $B_y$. Based on this, it was clear that we should consider the possible effects of IMF $B_y$ on the convection observed at the location of Cluster in our study. According to the model data we present (Fig. 5), there is definite evidence of IMF $B_y > 0$ penetration (locally), highlighted by the fact that the SC observed $B_y = 0$ when $B_x = 0$ prior to the flapping interval (lines 520-532). Whether this is governing the nature of the convection (locally), however, is another matter (discussed below in response to comment 2).

**Comment 2:** *Second, the IMF penetration and the polar convection are the process in global scale, while the dusk- dawn flow associated with current sheet in this case is at much less scale. So the analysis of IMF and polar convection can not support inconsistency between the expected By in current sheet and the observed By during the crossing of current sheet.*

**Response 2:** In our study, the flows observed by C1 during the flapping would have been consistent with IMF $B_y$ control *if* we had a situation where IMF $B_y < 0$ penetration had

occurred (lines 351-355). This motivates our reasoning for needing to look on a global scale, so we use the IMF and SuperDARN data to demonstrate what the sense of the large-scale magnetospheric asymmetry is. This data tells us what (if any) sense of IMF $B_y$ has penetrated into the magnetosphere, and conveys that it is definitely **not** IMF $B_y < 0$; in-fact, it is consistent with IMF $B_y > 0$. This is a critical detail, because it means that the observed dusk-dawn flow associated with the current sheet flapping is therefore definitely not IMF $B_y$ controlled. It is an important distinction that we **do not** use the IMF and polar convection data to interpret the dynamic behaviour of the plasma and magnetic field that is occurring in the current sheet. Instead, we separately examine whether the current sheet flapping might be responsible for driving the variable dusk-dawn flow. The negative $B_y$ perturbations observed by C1 during the flapping are consistent with perturbations in the dusk-dawn flow (lines 280-281), and are clearly unrelated to any IMF $B_y$-effect.

**Reviewer 2**

**Comment 1:** *Lines 585-645: My main concern is how well the curlometer current J and the JxB force can be used to describe this dynamical situation. Because of the large inter-spacecraft separation of the spacecraft, the estimates of these quantities are averages over a large volume.*

**Response 1:** We fully acknowledge that the separation of the spacecraft has implications for the interpretation of the curlometer analysis. As such, we very consciously draw only qualitative conclusions regarding the $J \times B$ control of the flows, in relation to how the measured estimates compare to modelled values derived from the expected background field. We are confident that the extent to which we interpret these estimates is reasonable. We are reassured through discussions with Malcolm Dunlop (Dunlop et al., curlometer technique papers, 1988, 2002; and as acknowledged in the manuscript) regarding our application of the curlometer technique in case of large inter-spacecraft separation. He confirmed that although large, the s/c configuration looks acceptable. The estimate should be stable, but as the reviewer rightly says, only of the average of a large volume.

**Comment 2:** *The flapping of the current sheet is observed only in one part of this volume. Should one compute these quantities specifically for C1 if that would be possible? If one assumes that the computed current J is stable and represents the current over the region covered by the Cluster tetrahedron, would it be reasonable to compute the JxB force using that J and then the B field measured only by C1? That would be a more local estimate for the JxB force at the C1 position. The authors could compute that and compare to the present estimate.*

**Response 2:** We agree that the flapping of the current sheet is only observed in one part of the tetrahedron volume, given that C1 is the only of the Cluster spacecraft to observe this. The reviewer puts forward an interesting and helpful suggestion to compute the $J \times B$ force

using $J$ (as calculated from all 4 SC), but using $B$ only measured by C1 (instead of averaging $B$ across the four SC). The results of this analysis are presented in Figure R1, where in panel vi) one may observe the results of calculating $J \times B$ using the measured magnetic field of C1 (solid line) and the modelled magnetic field of C1 (dashed line).

[Figure]

**Figure R1:** As in Figure 6 in the manuscript, but with $(J \times B)_y$ calculated using $B$ from C1 only.

Firstly, consistent with the original Figure 6, panel vi), as perhaps to be expected, the 'model' $J \times B$ force (dashed line, where $J$ has been calculated using the model C1 field and the true fields from C2-C4) is still weakly dawnward, consistent with the background 'curvature' of the magnetic field at the pre-midnight location as suggested in the original manuscript (note the different y-scale compared to the original panel vi). Particularly different, however, is the magnitude of the newly calculated $J \times B$ force. In the older

analysis, the calculated $\boldsymbol{J} \times \boldsymbol{B}$ force was always positive with respect to the model $\boldsymbol{J} \times \boldsymbol{B}$ force, but still net negative for most of the interval. In this newer analysis, the $\boldsymbol{J} \times \boldsymbol{B}$ force is still mostly positive with respect to the model $\boldsymbol{J} \times \boldsymbol{B}$ force but is now also mostly net positive. If one takes the original panel vi) and subtracts the model $(\boldsymbol{J} \times \boldsymbol{B})_y$ from $(\boldsymbol{J} \times \boldsymbol{B})_y$, the result would be similar to $(\boldsymbol{J} \times \boldsymbol{B})_y$ in the new analysis. This suggests that just using $\boldsymbol{B}$ from C1 in calculating $(\boldsymbol{J} \times \boldsymbol{B})_y$, rather than averaging across the four SC, has reduced the effects of the larger-scale background field curvature (incorporated by including the other SC). Compared with Fig. 6, the perturbations to $(\boldsymbol{J} \times \boldsymbol{B})_y$ are now also much larger in magnitude (previously they ranged between +/- 4 Nm$^{-3}$, but now they range between around -8 and 25 Nm$^{-3}$). We suggest that this is related to the fact that we are now using the (highly variable) magnetic field of C1 to provide the $\boldsymbol{B}$ in our computation of $\boldsymbol{J} \times \boldsymbol{B}$, as opposed to an averaged $\boldsymbol{B}$ – which, previously, was much steadier and closer to 0 in magnitude.

Previously, we drew attention to two key features of this figure. Firstly, we argued that the perturbations to $(\boldsymbol{J} \times \boldsymbol{B})_y$ were mostly associated with the magnetic field perturbations observed by C1. Inherently, this is even more apparent now. Secondly, we argued that 'the dawnward flow bursts tend to occur when $(\boldsymbol{J} \times \boldsymbol{B})_y$ is more negative, with the weak duskward flow bursts occurring when $(\boldsymbol{J} \times \boldsymbol{B})_y$ is less negative'. The new analysis suggests a slight adjustment to this interpretation is necessary. The dynamics evident in panels (vii) and (viii) now appear to be almost always associated with positive (duskward) enhancements in $(\boldsymbol{J} \times \boldsymbol{B})_y$, in contrast to the background (model) dawnward sense of $(\boldsymbol{J} \times \boldsymbol{B})_y$. This suggests that there is less variability in the direction of the $(\boldsymbol{J} \times \boldsymbol{B})_y$ perturbations, and that instead, the dynamic behaviour of $(\boldsymbol{J} \times \boldsymbol{B})_y$ is simply consistent with localised kinks in the magnetic field that are associated with the transient perturbations to the dawn-dusk flow. In fact, this is now more consistent with our "cartoon" interpretation presented in Fig. 7. Overall, this new version of the analysis does not alter our fundamental conclusions, in which we already acknowledge the uncertainty in trying to make detailed one-to-one association between the $(\boldsymbol{J} \times \boldsymbol{B})_y$ and flow perturbations. We consider that both approaches involve assumptions that limit the extent of the interpretation and conclude that a sensible way forward is to present both approximations in a new Fig. 6, and to highlight that the conclusions we have drawn are supported by both.

**Comment 3:** *Lines 525-532, Summary and Abstract, lines 106-107: Second, the typical extents of the IMF $B_y$ penetration that is overriding the tail field line flaring (and causing tail magnetic field line twisting) in the case of clearly nonzero IMF $B_y$ (IMF $|B_y| > 3$ nT) can be seen in Figure 2 of Pitkänen et al. (2019, GRL). Their Figure 2a and 2b show that under clearly positive IMF $B_y$ conditions, the (slow) earthward convection is expected to be on average duskward both above and below the neutral sheet at the position of the Cluster spacecraft of the present manuscript. The tail magnetic field in this position is expected to be governed by the flaring. In the case of the present manuscript, the magnitude of positive IMF $B_y$ was mostly less than +3 nT. Therefore, the global flow pattern in the magnetotail*

*could be assumed to be even less asymmetric and the tail field line twisting occur at smaller extents than in Figure 2 of Pitkänen et al. (2019). The Cluster magnetic field data (C2-C4 data) clearly demonstrate the appearance of the field line flaring in the case of the present manuscript and not the twisting of the field lines due to IMF $B_y$ influence. Furthermore, I think that while model results, Figure 5b in the present manuscript nicely demonstrates the spatial limits of the IMF $B_y$ penetration to twist the tail magnetic field lines. The authors could modify the Summary section and add there that in this event, the IMF $B_y$ influence in the position of Cluster was not strong enough to twist the magnetic field lines and the measured flows were associated with the localized magnetic field perturbation. So, the current sheet flapping was not overriding the IMF $B_y$ control, because the control did not exist at the location of Cluster. Also then the end of the abstract (and the text elsewhere where IMF $B_y$ overriding is discussed) would need to be modified.*

**Response 3:** We agree with the reviewer that Fig. 2 of Pitkänen et al. (2019) nicely illustrates the extents of the IMF $B_y$ penetration, and that at the location of the Cluster spacecraft, the convection is expected to have a duskward component both above and below the neutral sheet. Of course, in our study, this is not what we see; the flow observed by Cluster during the flapping interval tended to have a clear dawnward component in the southern hemisphere, in disagreement with the spatially-averaged picture of slower (< 200 kms$^{-1}$) flow presented by Pitkänen et al. (2019). The local $B_y$ that C1 observed during the flapping interval was also mostly negative (irrespective of hemisphere), also inconsistent with the average picture of Pitkänen et al. (2019).

We also agree with the reviewer that the C2-C4 data demonstrate field line flaring. However, we think that the suggestion that these observations do not indicate the presence of field line 'twisting' due to IMF $B_y$ influence is a bit ambiguous. If, by 'twisting', the reviewer means that this IMF $B_y > 0$ perturbation was unable to change the sign of the (expected) $B_y < 0$ field in the pre-midnight northern hemisphere (e.g. at the location of C2 and C4), then the reviewer is absolutely correct. However, as we noted in the manuscript (lines 225-227, 520-525), and in relation to the discussion of Fig. 5, the spacecraft observing $B_x = 0$ and $B_y = 0$ just prior to the flapping interval was, in itself, indicative of IMF $B_y > 0$ penetration.

In relation to the convection: the flows that Cluster observed (locally) could have been expected in the case of a situation where we had IMF $B_y < 0$ (lines 351-355). However, the IMF and SuperDARN data allowed us to confirm an absence of any IMF $B_y < 0$ (and in-fact, the large-scale picture was one which seemed consistent with IMF $B_y > 0$). As the reviewer therefore suggests, the observed flows must have been associated with the localized magnetic field perturbations in $B_y$ (lines 280-281) and the current sheet flapping, and could not be explained by IMF $B_y$ control. Clearly, any IMF $B_y > 0$ associated perturbation at the location of Cluster was not significant enough to control (or perhaps override the flapping-related control) the dusk-dawn flow. We therefore agree with the reviewer on this point, and have reworded where applicable in the manuscript: rather than the 'flapping overriding the IMF $B_y > 0$ control of the flow', we now suggest that the 'IMF $B_y > 0$ penetration at the

Cluster location was unable to override/overcome the variable dusk-dawn flow associated with the flapping'.

**Comment 4:** *Line 401: Maybe write here "the Harang reversal" instead of "the Harang discontinuity", because the authors are investigating flows.*

**Response 4:** We have amended this to 'Harang reversal' in the revised manuscript.

**Comment 5:** *Line 701: Which plasma sheet magnetic field observations the authors do mean here? The TA15 model results?*

**Response 5:** Here, we are referring to how e.g. prior to the flapping interval the SC tended to observe $B_y = 0$ at $B_x = 0$, which through our discussion related to Fig. 5 (the model results) we used to show was an effect of IMF $B_y > 0$ penetration. We have reworded this to be clearer: "The IMF, ionospheric convection, and comparison of the plasma sheet magnetic field observations to the TA15 model field, all lead to…"

**Additional Comment:** *In my opinion, it would be good to have both approximations for (**JxB**)$_y$ presented in Figure 6. That would give more information about the situation. Then highlighting that the conclusions the authors have drawn are supported by both of the approximations would be fine.*

**Response:** We thank the reviewer for their suggestion and have included this in the revised manuscript.

**Reviewer 3**

**Comment 1:** *I therefore recommend that the paper is restructured to reduce the early discussion of asymmetric tail untwisting, and not to interpret the observations as a departure from that (since the spacecraft seem to be in a location where the tail By component is dominated by flaring, rather than IMF penetration), but instead to frame the interpretation in terms of the negative Vperp_y values departing from the expected duskward convection at this location.*

**Response 1:** We agree that the expected flow at the pre-midnight location of the spacecraft is duskward, and that the departure from this, in the southern hemisphere observations in particular, is the main inconsistency between what is 'expected' and what is seen. However, we argue that the pre-midnight location does not necessarily preclude dawnward flow in the southern hemisphere in the case that a strong IMF $B_y < 0$ twist is present. Critically,

previous studies (such as those by Pitkänen et al., 2013, 2017) have based their interpretation of similar fast flows on this very assertion. So, we think that it is still important to rule out the possibility that C1 is actually observing tail untwisting due to IMF $B_y < 0$ penetration. We have attempted some restructuring along the lines suggested, including mentioning the expected duskward convection at the spacecraft location (even in the case of no large-scale asymmetry) and brought in the concept of magnetotail flaring much earlier in the manuscript.

**Comment 2:** *Lines 65-8: This is the predominant behaviour, but it is location dependent (i.e. flaring dominates away from midnight).*

**Response 2:** We agree with the reviewer that the flaring effect will dominate further away from midnight. As noted above, previous studies such as Pitkänen et al. (2013, 2017) have investigated IMF $B_y$ control of magnetotail flows at up to ~ 7 $R_E$ towards the dusk-dawn flanks which revealed a clear dependence of the flows on IMF $B_y$, so we do think it is important to at least mention this here. However, have made clearer that this behaviour is expected to be dominant close to midnight, but that other sources of $B_y$ (away from midnight), such as flaring, are expected to be significant.

**Comment 3:** *Line 218: This is the first sign that flaring may be dominant, as I think the By sign reversal here is not what is expected in the tail twist scenario (near to midnight)? Similarly for the observations described at lines 223-5*

**Response 3:** We agree with the reviewer that this suggests that the flaring is dominant, both at line 218 and lines 223-225. We have made it clearer at these points in the manuscript that these observations are consistent with magnetotail flaring. The reviewer is also correct in that closer to midnight, one might expect to observe $B_y > 0$ irrespective of hemisphere (see Fig. 5b), in the case of IMF $B_y > 0$ tail twisting.

**Comment 4:** *Line 220: I think it is important to mention in the text that the solid lines in Fig 2b iv-vi are the field-perpendicular component, and the dotted lines are the total velocity components. This information is in the figure caption, but it only becomes apparent in the main body of the text at line 242.*

**Response 4:** We have amended this in the revised manuscript.

**Comment 5:** *Lines 258-62: Here, you are again describing observations that are consistent with flaring.*

**Response 5:** Here, does the reviewer mean 'inconsistent' with flaring? In the pre-midnight sector, one would expect to observe $B_y > 0$ where $B_x < 0$ due to the flaring. Instead, C1 observes $B_y < 0$. We do allude to the inconsistency with flaring on line 260: '…this is inconsistent with what we would expect based on the location of the spacecraft…'. Of

course, this is also unrelated to any IMF $B_y$-effect and is instead related to the presence of a localised perturbation.

**Comment 6:** *Line 278: I think it might be worth rewording this slightly, as the periods of positive Bx also include observations of Vperp_y that are close to zero or even negative (particularly from 00:30-00:31 UT).*

**Response 6:** We think generally, positive Vperp_y is observed when positive $B_x$ is observed (e.g. 00:30:00 UT, 00:31:20 UT, 00:31:40 UT), but we agree with the reviewer that particularly from 00:30 – 00:31 UT, a mix of positive and (weakly) negative Vperp_y is observed when C1 measures positive $B_x$. We have tweaked this statement accordingly in the revised manuscript to be less definite, e.g. 'At times when $B_x$ became positive, indicating that C1 was above the neutral sheet, C1 observed positive (duskward) $v_{\perp y}$ a majority of the time, although this flow barely reached 100 km s$^{-1}$.'

**Comment 7:** *Line 280: To my eye, the positive enhancements in Vperp_y do not seem to be associated with negative enhancements in By. They mostly seem to be associated with either no particular By signature, or a reduction in negative By or positive By turning.*

**Response 7:** We think the reviewer is correct here. Perhaps, the only exception to this is at ~00:30 UT, where there is a clear decrease in $B_y$ in association with the positive Vperp_y enhancement. We think this is easily remedied by changing our statement on line 280 to: 'The negative enhancements in $v_{\perp y}$ were generally accompanied by negative enhancements in $B_y$', as this is clearly much more apparent.

**Comment 8:** *Lines 357-9: Emphasising the expectation from IMF penetration here seems inappropriate, as the observations so far seem to establish that tail flaring is the dominant source of By at this location.*

**Response 8:** It is important to highlight that our assertion here (lines 357-359) is based *solely* on the IMF data. We do agree with the reviewer that the observations have shown that the flaring is dominant at this location. However, given that previous studies (Pitkänen et al., 2013) have shown an IMF $B_y$-effect on convection to exist at this location, we think it is important to at least consider. Of course, what we go on to show is that the IMF and ionospheric convection observations are not consistent with a negative tail $B_y$, instead pointing towards there being a large-scale IMF $B_y > 0$ asymmetry. Certainly they reveal the absence of any IMF $B_y < 0$ effects, which could have explained the dawnward flow observed by C1 in the southern hemisphere.

**Comment 9:** *Lines 420-3: I was confused by this sentence, as surely even when untwisting happens, the convection cell to which a spacecraft is connected also depends on its local time? Even with untwisting, there are two convection cells (i.e. some field lines return via dawn, and others via dusk), it's just they're asymmetric.*

**Response 9:** The reviewer is correct that in actuality, the flows that a spacecraft is expected to observe in association with the untwisting are dependent on MLT. However, in the study of Pitkänen et al. (2013), when considering e.g. IMF $B_y > 0$, in the northern hemisphere, only

a dawnward flow (in association with the extended dawn cell) would be counted as a flow which agrees with the untwisting hypothesis. A duskward flow in the northern hemisphere, meanwhile, would have been considered to be a flow which disagrees with the hypothesis. Clearly, this is problematic, as it may simply be the case that the spacecraft is located pre-midnight and is observing return duskward convection associated with the dusk cell. This is why our attention is focused on the southern hemisphere, where the observed pre-midnight dawnward flow could only feasibly be explained by a strongly negatively (IMF $B_y <$ 0) twisted tail. This is again why we think is important to address and rule out the possibility of an IMF $B_y$ effect. Consequently, the evidence of a large-scale IMF $B_y > 0$ asymmetry is clearly inconsistent with the observed dawnward flow in the SH – which we instead suggest is associated with the flapping current sheet.

**Comment 10:** *Lines 440-2: I don't think this statement is correct. The northern hemisphere footprints map close to the boundary between the dusk and dawn cells, and the lack of scatter at the northern hemisphere footprint makes it hard to be specific about which convection cell the footprints actually lie in. The authors seem to acknowledge this as a possibility in the lines that follow, but I think the sentence here is too definite. I do agree, though, that the duskward flow seen at the southern hemisphere footprint conflicts with the generally negative Vperp_y observed by C1 in Figure 3 (though it agrees with the generally positive Vperp_y observed by C3, and also by C1 earlier, in Figure 2).*

**Response 10:** In terms of the specific map that we are referring to (00:30 – 00:32 UT), we do think that the spacecraft footpoints appear to map closer to the dawn cell. But we agree that this statement may be too definite, so have tweaked it slightly to: 'the spacecraft appear to map closer to the dawn cell than the dusk cell, such that the predominantly duskward flow that C1 observed in the northern hemisphere plasma sheet would seem to be inconsistent'.

**Comment 11:** *Lines 464-8: Just to note that the duskward flow observed by C3 is also consistent with the duskward convection that would be expected in the absence of tail twisting, given the spacecraft location. But I agree with the statement on lines 468-72 that the difference between C3 and C1 means something more local is happening at C1*

**Response 11:** We agree with the reviewer on this point, and we have made this clearer in the revised manuscript: '…;although, it should be noted that this observation would also be consistent with the expected duskward flow in a pre-midnight location even in the absence of a large-scale asymmetry (e.g. Kissinger et al., 2012).'

**Comment 12:** *Lines 477-8: Since C2 and C4 are in the northern hemisphere (from Bx), their negative By seems to be consistent with flaring.*

**Response 12:** We agree with the reviewer here. This is alluded to on lines 528-532, but we have clarified this earlier in the revised manuscript (e.g. lines 256-259)

**Comment 13:** *Line 537: I think the word "crossing" is superfluous here.*

**Response 13:** We are grateful to the reviewer for pointing this out. We have removed 'crossing' from the revised manuscript

**Comment 14:** *Line 696: Do the authors mean Figure 7d?*

**Response 14:** Here, we are referring to Figure 6d – the schematic in Figure 7d is trying to depict a 'snapshot' which encapsulates the physics & observations across the time window (green shaded region in Fig. 6d).

**Comment 15:** *- Point 1 (lines 700-2) says that the IMF, ionospheric convection and plasma sheet observations all lead to the expectation of an IMF $B_y > 0$ asymmetry. This is true (with respect to the IMF and ionospheric convection) on a global scale, and indeed the ionospheric observations do show asymmetric flows across midnight. But I feel the sentence is a bit misleading, as the IMF and ionospheric observations do not give us grounds to suggest we observe distinct tail untwisting at the location of Cluster (because the northern hemisphere footprint cannot be confidently placed on the dawn cell, given its proximity to the dusk cell and the lack of local scatter, and the southern hemisphere footprint is at a location that would observe duskward flow even without untwisting). Furthermore, I think the sentence is incorrect in saying that the plasma sheet observations show a large-scale asymmetry - the magnetic field observed by C2/C3/C4 (and also C1 before the flapping) seem entirely consistent with flaring and do not show evidence of the penetrated By component being dominant here. Likewise, the convective flows observed by C3 (and C1 before the flapping) are duskward, consistent with the spacecraft being far enough from midnight that flaring is the dominant cause of By, and hence the field lines convect sunward in the same sense as they would in the symmetrical case.*

**Response 15:** There are a couple of points raised in this comment, and so we address each one in turn (reviewer comment in italic):

*I feel the sentence is a bit misleading, as the IMF and ionospheric observations do not give us grounds to suggest we observe distinct tail untwisting at the location of Cluster (because the northern hemisphere footprint cannot be confidently placed on the dawn cell, given its proximity to the dusk cell and the lack of local scatter, and the southern hemisphere footprint is at a location that would observe duskward flow even without untwisting)*

We agree with the reviewer that the IMF and ionospheric observations do not allow us to suggest that we are observing tail untwisting at the location of Cluster. What the IMF and ionospheric data do allow us to provide evidence of, however, is a large-scale IMF $B_y > 0$ asymmetry (and a clear absence of any IMF $B_y < 0$). This is a critical detail, because if the global-scale observations had revealed a strongly negatively (IMF $B_y < 0$) twisted tail, then the C1 observations *would* have been consistent with tail untwisting. This is why it is

particularly important for us to rule out this possibility. The fact they were not consistent with this, however, and instead appeared to be associated with the flapping of the current sheet, means that we cannot be observing flow associated with tail untwisting at C1. In terms of C3, the reviewer is correct that the observed duskward flow in the southern hemisphere could simply be consistent with the spacecraft location.

*Furthermore, I think the sentence is incorrect in saying that the plasma sheet observations show a large-scale asymmetry - the magnetic field observed by C2/C3/C4 (and also C1 before the flapping) seem entirely consistent with flaring and do not show evidence of the penetrated By component being dominant here. Likewise, the convective flows observed by C3 (and C1 before the flapping) are duskward, consistent with the spacecraft being far enough from midnight that flaring is the dominant cause of By, and hence the field lines convect sunward in the same sense as they would in the symmetrical case.*

We agree that we need to be clearer on this point. We are not suggesting that the plasma sheet magnetic field observations show evidence of the penetrated $B_y$ component being 'dominant' over the flaring – we simply argue that (combined with the IMF and ionospheric data) they lead to the expectation of a large-scale IMF $B_y > 0$ asymmetry. By comparing the plasma sheet magnetic field data to the TA15 model data, the fact that the spacecraft observed coincident $B_x$, $B_y = 0$ was, in itself, evidence of 'weak' IMF $B_y > 0$ penetration. This is evidenced in Fig. 5, which shows that in the absence of an IMF $B_y$ penetration (left panel) we ought to see a negative $B_y$ when Cluster crosses the neutral sheet. In order to see zero $B_y$ at the crossing points the right hand panel suggests that a weak positive IMF $B_y$ must have penetrated. We have re-worded this in the manuscript to be clearer: 'The IMF, ionospheric convection, and comparison of the plasma sheet magnetic field observations to the TA15 model field, all lead to…'. But, as above, we agree with the reviewer that the convective flows observed by C3/C1 prior to the flapping are consistent with the spacecraft location.

***Comment 16:** - I think that referring to magnetotail untwisting specifically in point 2 (lines 702-6) is not justified, because of the fact that IMF penetration does not seem to be the main cause of the By components observed at the location of Cluster. But this is easily remedied by reducing the emphasis on tail twisting, and instead comparing with the duskward convection that is expected (and observed, by C3) at this location.*

**Response 16:** In our study, the IMF $B_y$ had been generally positive for many hours prior to the interval. Previous studies (Pitkänen et al., 2013, 2015) have interpreted their spacecraft observations in the context of magnetotail untwisting (even up to $Y_{GSM} \sim 7 R_E$), which is why we too frame our observations in this context. We do agree that flaring is dominant over the IMF $B_y$ -effect. However, the dawnward flows in SH are inconsistent with expected duskward convection at the C1 location but *would* be consistent with a strong IMF $B_y < 0$ twist. This is why we think that it is important to rule that the possibility of any IMF $B_y < 0$ penetration in the manner that we do in our study. We have attempted to reduce the emphasis on this and introduced the concept of flaring much earlier in the paper.

*Comment 17:* - *I agree with point 3 (lines 706-8), but this is really with respect to the general duskward convection that would be expected at this location, given the local time, TA15 modelling, and the fact that Cluster observes predominantly By components consistent with flaring. I see no reason why, given the authors' results, similar local processes should not be observed nearer to midnight, and therefore act against the tail untwisting process, but I think that is a matter for suggestion for a future study.*

**Response 17:** We agree with the reviewer on these points. We note that a statistical study on the dusk-dawn magnetotail flows is underway, in which we consider a range of Y-locations.

*Comment 18:* *Figures (general): There is a standard colour legend for Cluster line traces, which does make it easier to keep track of which trace corresponds to which spacecraft. I would encourage the authors to use that in those figures with Cluster data or footprints, as it really does aid a reader who is familiar with Cluster data. Some of the figures would benefit from being larger.*

**Response 18:** We have now used the standard colour legend for Cluster line traces in the revised manuscript. It's not clear which figures the reviewer was referring to as being too small – all the figures are a full page width, so we are not sure how to address this point.

*Comment 19:* *Figure 6: The y axis labels are somewhat crowded in some panels. Also, the green boxes (a) and (c) don't quite line up with the features that are discussed in the text - I think the features being highlighted here are the negative excursions in By, so I think both boxes should move a little to the right?*

**Response 19:** We thank the reviewer for pointing this out. We have tidied up the y-axis labels and moved the boxes a little to the right.

---

## Author Response (AR2)

We would once again like to thank both of the reviewers for their careful consideration of the manuscript and their very helpful comments. Our responses are provided below.

**Reviewer 1**

*Comment 1: One further comment has arisen when reading the revised manuscript. In the first sentence of the abstract (lines 12-14 in the tracked-changes revised version), the authors give a strong statement for the dusk-dawn sense of both slow (< 200 km/s) and fast (> 200 km/s) convective magnetotail flows is strongly governed by IMF By conditions. The authors apparently are referring to Pitkänen et al. (2019) and Pitkänen et al. (2013; 2017) papers here. I would like to suggest the authors to ease the sentence a bit to the following form: "Previous observations have provided a clear indication that the dusk-dawn (Vperpy) sense of both slow (< 200 km/s) and fast (> 200 km/s) convective magnetotail flows can be governed by the interplanetary magnetic field (IMF) By conditions." This is because it is good to keep in mind that the slow flow patterns in Pitkänen et al. (2019) are average patterns and in individual situations the coverage of the flows with the dominating Vperpy direction is expected to vary. They are also from the data with clearly nonzero IMF By. In the results for fast flows (Pitkänen et al., 2013; 2017), even when taking only the subset of events with mean tail By over the event collinear to IMF By, there still appears to be some fast flows in each category with mean Vperpy to an opposite direction than expected from the untwisting hypothesis/model. Also, there can be flow events in which the mean Vperpy direction formally agrees with untwisting hypothesis, but these flows occur in a region where the flaring dominates and screens the IMF By penetration and might occur also without IMF By penetration. However, these flows are still counted as agreeing with the untwisting hypothesis.*

**Response 1:** We agree with the reviewer and would like to thank them for their helpful suggestion. We have amended '*strongly governed..*' to '***can be** governed..*' in the revised manuscript. We note that we have also tweaked the abstract further, in reference to Reviewer 2's first comment, below.

**Reviewer 2**

*Comment 1: Abstract: The whole abstract is about the importance of IMF control/untwisting and how these observations are inconsistent with this untwisting hypothesis. There is no mention of the fact that the twisting effect does not seem to extend as far as Cluster for this event (from the background flaring field), or the fact that the C1 flows (at least those south of the neutral sheet) are also inconsistent with the symmetric return direction that would be expected here. You suggest at line 27 that IMF penetration at the location of Cluster was*

*unable to override the variable flow associated with the flapping, but surely the point is that the flapping overcomes the \*net\* return direction, which in this case, the data suggest, is predominantly the symmetric element due to the location of the spacecraft.*

**Response 1:** We are grateful to the reviewer for pointing this out. In the abstract, we have now further emphasised that at the spacecraft location, the expected flow direction is duskward, unless there is untwisting due to a negative IMF $B_y$. Our discussion of the $B_y$ penetration then serves only to rule out this possibility. We conclude by saying, as the reviewer suggests, that the flapping overcomes the net duskward direction of the large-scale flow that is expected at the spacecraft location.

***Comment 2:*** *Introduction: The discussion around lines 50-83 is good and more balanced, but the end of the introduction (lines 107-116) says that the interval provides an opportunity to investigate the competition between IMF By control and localised dynamics, that the observed dusk-dawn direction of the (C1) transient flows disagrees with what would be expected from prevailing IMF By conditions, and concludes that IMF By penetration was unable to overcome the variable dusk-dawn flow associated with the flapping. Surely, the message that should be being conveyed here is that there is global evidence of untwisting (SuperDARN), but the C2/3/4 observations show that the spacecraft are outside the region where IMF control dominates (i.e. the large-scale return flows here are "symmetric"), but that this global pattern is overcome by a variable dusk-dawn flow associated with the flapping?*

**Response 2:** We thank the reviewer for pointing this out and agree with their suggestion about the message that this paragraph should convey. We have now re-worded the paragraph to be much more in-line with this: '*In this paper we present Cluster spacecraft observations of an interval of dynamic magnetotail behaviour on 12 October 2006, **prior to which the $B_y$ component of the concurrent upstream IMF had been largely positive for several hours**. Throughout this interval, Cluster 1 observed oscillations in the magnetic field $B_x$ component, which we attribute to current sheet flapping, concurrent with a series of convective fast flows with significant and variable dusk-dawn components. **'Observations from Cluster 2, 3 and 4 indicated that the spacecraft were at a pre-midnight location where magnetotail flaring was dominating over IMF $B_y$ control of the flows, resulting in the expectation of (symmetrical) duskward return flows (Pitkänen et al., 2019). In the southern hemisphere, such duskward flow was measured by Cluster 3, but not observed by Cluster 1, which instead measured flows with significant dawnward components. These dawnward flows were therefore inconsistent with any expectation that the flow was governed by flaring and, owing to evidence of large-scale IMF $B_y > 0$ ionospheric convection pattern, could also not be explained by the magnetotail untwisting hypothesis. We instead suggest that the current sheet flapping was exciting the variable dusk-dawn flow, overriding the expected large-scale duskward convection at the location of Cluster 1.*'

***Comment 3:*** *Line 349: "we discuss our rationale for interpreting the flows as being inconsistent with large-scale magnetotail untwisting and our interpretation of their relationship to current sheet flapping" - the emphasis and lack of acknowledgement of the symmetric element again seems odd given that the C2/3/4 magnetic fields are consistent with being dominated by flaring, and whilst the the weak background flow observed in the southern hemisphere (C3) is consistent with either the symmetric element or untwisting, there's nothing to rebut the assumption (from the lack of IMF control dominance in the B field observed by Cluster) of a background duskward convection (which is inconsistent with the C1 observations south of the neutral sheet).*

**Response 3:** The reviewer makes a valid point here. We have tweaked this statement to: '*we discuss our rationale for interpreting the flows **observed by C1** as being inconsistent with **the large-scale convection expected based on the spacecraft location and** magnetotail untwisting **considerations,** and our **alternative** interpretation of their relationship to current sheet flapping.*'. So our new statement emphasises that it is the C1 observations we are referring to, and the first point we make in the above sentence is that the observed flows are inconsistent with the (symmetric) large-scale convection expected at the pre-midnight location, which we now expand on at the beginning of Section 4.1.

***Comment 4:*** *Section 4.1: You first seek to rule out the magnetotail untwisting hypothesis, but given the C2/3/4 observations, surely the first point to make is that the C1 flow is inconsistent with the expected symmetric return flow direction (which is seen by C3), and \*then\* that it cannot be explained either by IMF control. Fundamentally, given the C1/3/4 observations and TA15 modelling, untwisting/IMF control at this location seems like an odd choice of null hypothesis.*

**Response 4:** We agree with the reviewer about ensuring that the first argument we should make is how C1's observations are inconsistent with the 'symmetric' flow picture. We have therefore amended the first paragraph of Section 4.1 to read as follows: '*During the five-minute interval studied (00:28 – 00:33 UT) C1 measured a continually fluctuating $B_x$ component (Fig. 3i), indicative of multiple crossings of the tail current sheet. C1 was the only spacecraft to measure this signature across the interval (although similar signatures had been observed a few minutes earlier by C2 and C4). C1 also measured a series of earthward convective magnetotail fast flows with varying dusk-dawn components. The data in Fig. 3 i) and Fig. 3 v) illustrate that when $B_x$ was positive (negative), a duskward (dawnward) $v_{\perp y}$ was generally observed. **The observed dawnward flow in the southern hemisphere, in particular, is inconsistent with the expected symmetric duskward flow at the pre-midnight location of C1 which was, however, observed by C3. This suggests that the typical 'symmetrical' Dungey-cycle return flow (e.g. Kissinger et al., 2012) cannot provide an explanation for the flow observations made by C1. We thus turn our attention to other possible explanations which we explore in detail, below.*'**

***Comment 5:*** *SuperDARN observations (Lines 444-470): I agree that the SuperDARN observations show that there is a large-scale asymmetry in the flow, consistent with the*

*expected IMF control, but I reiterate my concern that they do not convincingly show that the Cluster spacecraft are located on the asymmetric part of the flow. The authors state in their response that "In terms of the specific map that we are referring to (00:30 – 00:32 UT), we do think that the spacecraft footpoints appear to map closer to the dawn cell", and have made their statement slightly less definite; they go on to acknowledge that the proximity to the dusk cell and uncertainty in the field line trace "may give credence to the possibility" that the spacecraft map to the dusk cell. However, I would point out that there is a large region devoid of scatter near the footprints of the spacecraft; dawnward and duskward flow are observed dawnward and duskward of the spacecraft, respectively, but if the conclusion (that Cluster is nearer the dawn cell) is based on the backscatter, then there is insufficient scatter to conclude exactly where the division between the two return directions occurs based on the scatter alone. If the conclusion is based on the electrostatic potential contours, then it's notable that both 4 minutes earlier (at 00:24-00:26 UT) and 2 minutes later (00:32-00:34 UT) the footprints are closer to the dusk cell (i.e. the outermost solid contour) than the dawn cell (outermost dashed contour). I can't see any significant differences in the plotted data points where there is scatter at both of these times, but notably there is less scatter just poleward/dawnward of Cluster at 00:30-00:32 compared with 00:24-00:26 UT, and I suspect this will have affected the details of the interface between the dawn and dusk cells near the Cluster footprints, i.e. that the contours are highly sensitive to the absence of scatter in this region. Therefore, the northern hemisphere SuperDARN data are inconclusive on whether the spacecraft north of the neutral sheet are on the dawn or dusk cell, and certainly do not override the observational fact that C2/3/4 observe a flaring-dominated field and hence we would expect the return flow to be duskward (i.e. dominated by the symmetric element). Consequently the strong emphasis in this paragraph on untwisting being important at the location of Cluster is misleading. (NB The point about C1 not being on the dawn cell in the southern hemisphere (lines 462-464) is convincing, but again fully consistent with being part of the "symmetric" element.)*

**Response 5:** On reflection, we completely agree with the reviewers point about the northern hemisphere SuperDARN data being inconclusive, in terms of whether the spacecraft north of the neutral sheet map to the dawn or dusk ionospheric convection cell. As a result, we have completely reworded the first section of this paragraph when discussing the northern hemisphere convection maps: '***Firstly, let us consider** the northern hemisphere map from 00:24 – 00:26 UT in Fig. 4a: despite the lack of scatter in the immediate vicinity of the spacecraft footpoints, it is noticeable how the spacecraft appear to map closer to the dusk cell than the dawn cell. For the remaining northern hemisphere maps, there is insufficient scatter to determine the exact division between the dusk and dawn convection cells, such that it is inconclusive as to which cell the Cluster spacecraft map to when above the neutral sheet. If Cluster did indeed map to the dusk convection cell, then* the **duskward flows in the** northern hemisphere **plasma sheet** observed by C1 *would actually be consistent with the large-scale convection pattern*. **Furthermore, given that the C2-C4 magnetic field observations are consistent with the local $B_y$ being dominated by magnetotail flaring (as opposed to IMF $B_y$) at the pre-midnight location of Cluster, it is likely that we would expect the return sense of the convection to be dominated here by the symmetric (duskward) element both** above and below the neutral sheet *(see e.g. Pitkänen et al., 2019).*'

***Comment 6:*** *Section 4.2: I suggest wording lines 478-482 is a more even-handed way.*

**Response 6:** We have reworded these lines in a more balanced manner: '*The low-level of flow seen by C3 is mostly duskward (Fig. 3v),* **which would be** *consistent with untwisting* **for** *IMF $B_y > 0$,* **given its southern hemisphere location**. **We note, however, that due to the pre-midnight location of C3, one would also rightly** *expect* **to observe** *duskward flow* **even** *in the* **case that there was no IMF $B_y > 0$ control (e.g. Kissinger et al., 2012)'**.

***Comment 7:*** *Section 4.4: Bullet point 2 (lines 724-727) is very loaded to tail untwisting, but the fact is that the C1 observations are also not consistent with symmetric return flow either.*

**Response 7:** We have added more emphasis to the symmetric return flow element. This point now reads: '*The Cluster 1 spacecraft observed convective flow with a dusk-dawn component that was inconsistent with current theories of IMF $B_y$-induced dusk-dawn flows associated with magnetotail untwisting.* **Notably, the observed dawnward flow in the southern hemisphere, whilst inconsistent with IMF $B_y > 0$, was also inconsistent with the expected (symmetric) duskward flow at this pre-midnight location even in the absence of IMF $B_y$ control.'**. In addition, we have added a statement into the third point: 3) Magnetic field perturbations that were indicative of a localized current sheet flapping and dusk-dawn kink in the field occurred coincident with the flows. It therefore seems likely that in this case the IMF $B_y$-driven asymmetry, **or indeed the symmetric flow expected at the spacecraft location**, was **being overridden by** the localized dynamics in governing the dusk-dawn component of the flow.

***Comment 8:*** *Summary: I think you need a bullet point summarising the C2/3/4 observations, i.e. the magnetic field observations revealed a background field, local to Cluster, where the By component was dominated by flaring, indicating that the region of IMF control dominance did not extend as far as Cluster, and C3 showed southern hemisphere return flow that was duskward (inconclusive, but fully consistent with the symmetric element of return flows). This may need some reconciling with your SuperDARN bullet point.*

**Response 8:** We agree with the reviewer that it is important for us to summarise the observations from the other Cluster spacecraft and note how these are consistent with $B_y$ being dominated by flaring. Therefore, we have added a fourth bullet point which clarifies this: '***The C2, C3 and C4 magnetic field observations suggested that the local $B_y$ was being dominated by magnetotail flaring, as opposed to IMF $B_y$. C3 also observed duskward flow in the southern magnetic hemisphere, consistent with the symmetric flow expected owing to the pre-midnight location of the spacecraft.***'. We have also amended our second bullet point.

**Comment 9:** *Line 50: "flows are generally expected to be symmetric...at least in the absence of any asymmetry" - this is a truism, so perhaps reword!*

**Response 9:** We have removed the latter part of this sentence, so that it now just reads: *'Magnetotail flows are generally expected to be symmetric about midnight (e.g. Kissinger et al., 2012).'*

**Comment 10:** *Line 77: "at up to 7 Re towards the dusk-dawn flanks" - I suggest this could also be more clearly worded, e.g. "at |Ygsm| values up to 7 Re"*

**Response 10:** We have reworded this as suggested by the reviewer. The sentence now reads: *'...as well as during more transient, dynamic BBF-like intervals (Grocott et al., 2007) at $|Y_{GSM}|$ values up to 7 $R_E$ (Pitkänen et al., 2013).'*

**Comment 11:** *Line 274: It might help the reader to point out that this observation (By steadily negative for C2/4 and steadily positive for C3) is consistent with the larger-scale By here being dominated by flaring.*

**Response 11:** We have now added an additional sentence to clarify this: *'**These observations are consistent with the larger-scale $B_y$ at the spacecraft location being dominated by magnetotail flaring'***

**Comment 12:** *Line 572: "This flapping must be highly localized" - or alternatively, it could be low in amplitude (i.e. as far as we know, this could be global in terms of extent across the tail, but not large enough in amplitude to reach the other spacecraft).*

**Response 12:** The reviewer makes a very good point here. We have tweaked this statement accordingly: *'This flapping must be **either** highly localized **or low in amplitude**, as at the...'*

---

## Author Response (AR3)

Author's Response

As well as thanking both of the reviewers for their ongoing careful consideration of the manuscript, we would like to note that in the final version of the paper, as suggested, we have taken into account Reviewer 2's final comment. We have simply removed the suggestion that the spacecraft appear to map closer to the dusk cell, and rather than focus on a particular convection map, made a slight tweak such that this now reads: "*…in Fig. 4a and 4b, respectively. For the northern hemisphere maps, there appears to be insufficient scatter to determine the exact division between the dusk and dawn convection cells, such that it is inconclusive as to which cell the Cluster spacecraft map to when above the neutral sheet. If Cluster in-fact mapped to the dusk convection cell, however, then the duskward flows…*".